# Pathogen invasion-dependent tissue reservoirs and plasmid-encoded antibiotic degradation boost plasmid spread in the gut

Erik Bakkeren[1]*[†‡], Joana Anuschka Herter[1], Jana Sanne Huisman[2,3], Yves Steiger[1], Ersin Gül[1], Joshua Patrick Mark Newson[1], Alexander Oliver Brachmann[1], Jörn Piel[1], Roland Regoes[3], Sebastian Bonhoeffer[3], Médéric Diard[4,5], Wolf-Dietrich Hardt[1]*

[1]Institute of Microbiology, Department of Biology, ETH Zurich, Zurich, Switzerland; [2]Swiss Institute of Bioinformatics, Lausanne, Switzerland; [3]Institute of Integrative Biology, Department of Environmental Systems Science, ETH Zurich, Zurich, Switzerland; [4]Botnar Research Centre for Child Health, Basel, Switzerland; [5]Biozentrum, University of Basel, Basel, Switzerland

**\*For correspondence:**
erik.bakkeren@zoo.ox.ac.uk (EB);
wolf-dietrich.hardt@micro.biol.
ethz.ch (W-DH)

**Present address:** [†]Department of Biochemistry, University of Oxford, Oxford, United Kingdom; [‡]Department of Zoology, University of Oxford, Oxford, United Kingdom

**Competing interest:** The authors declare that no competing interests exist.

**Abstract** Many plasmids encode antibiotic resistance genes. Through conjugation, plasmids can be rapidly disseminated. Previous work identified gut luminal donor/recipient blooms and tissue-lodged plasmid-bearing persister cells of the enteric pathogen *Salmonella enterica* serovar Typhimurium (*S*.Tm) that survive antibiotic therapy in host tissues, as factors promoting plasmid dissemination among Enterobacteriaceae. However, the buildup of tissue reservoirs and their contribution to plasmid spread await experimental demonstration. Here, we asked if re-seeding-plasmid acquisition-invasion cycles by *S*.Tm could serve to diversify tissue-lodged plasmid reservoirs, and thereby promote plasmid spread. Starting with intraperitoneal mouse infections, we demonstrate that *S*.Tm cells re-seeding the gut lumen initiate clonal expansion. Extended spectrum beta-lactamase (ESBL) plasmid-encoded gut luminal antibiotic degradation by donors can foster recipient survival under beta-lactam antibiotic treatment, enhancing transconjugant formation upon re-seeding. *S*.Tm transconjugants can subsequently re-enter host tissues introducing the new plasmid into the tissue-lodged reservoir. Population dynamics analyses pinpoint recipient migration into the gut lumen as rate-limiting for plasmid transfer dynamics in our model. Priority effects may be a limiting factor for reservoir formation in host tissues. Overall, our proof-of-principle data indicates that luminal antibiotic degradation and shuttling between the gut lumen and tissue-resident reservoirs can promote the accumulation and spread of plasmids within a host over time.

## Editor's evaluation

This work reveals an important feature of within-host acquisition and spread of antibiotic resistance. Focusing on a pathogen that shuttles between the gut lumen and tissue reservoirs, this study found that antibiotic degradation in the gut by resistant bacteria can promote the accumulation and spread of plasmids. This manuscript will be of interest to readers in the fields of infection biology, plasmid ecology, gut microbiomes, and antimicrobial resistance.

## Introduction

The accumulation of antibiotic resistance genes in pathogenic bacterial strains is an important cause of antibiotic treatment failure. Conjugative antibiotic resistance plasmids are key drivers of this accumulation, accelerating the emergence of new bacterial strains genetically resistant to antibiotics (*Wright, 2007*). The mechanisms that contribute to the spread of resistance plasmids within an infected host are still not fully established.

Recently, we have begun assessing the spread of plasmids within bacteria colonizing or infecting mammalian hosts. In this work, when we refer to 'host', we consistently refer to the mammalian host harbouring enteric bacteria. Plasmid spread within such hosts can be aided by tissue-lodged reservoirs of bacterial cells phenotypically recalcitrant to antibiotic therapy (*Bakkeren et al., 2019*). This process is called persistence and refers to a property of bacterial populations in which one subpopulation is killed by an antibiotic parallel to another subpopulation that is killed slowly ('persisters'), defined by phenotypic and not genotypic properties (*Balaban et al., 2019*; *Gollan et al., 2019*). Persisters can not only lead to antibiotic treatment failure, but they also have important implications for the evolution of antibiotic resistance (*Levin-Reisman et al., 2017*; *Liu et al., 2020*; *Windels et al., 2019*) or virulence (*Bakkeren et al., 2020*; *Diard et al., 2014*), and serve as long-term reservoirs promoting the spread of resistance plasmids. We showed this using the model organism *S. enterica* serovar Typhimurium (*S.*Tm) (*Bakkeren et al., 2019*). Invasive *S.*Tm cells form reservoirs of recalcitrant cells inside of host tissues (i.e. tissue persister reservoirs) that are difficult to treat with antibiotics (*Bakkeren et al., 2019*; *Claudi et al., 2014*; *Diard et al., 2014*; *Helaine et al., 2014*; *Kaiser et al., 2014*; *Stapels et al., 2018*). After treatment, these *S.*Tm cells can re-seed the gut lumen from their reservoirs along with any plasmids they carry. In the gut lumen, this re-seeding provides the plasmids with the opportunity to conjugate into new bacteria. Thereby, plasmids can subvert persisters to ensure long-term association within a given host. Besides persisters, *S.*Tm tissue invasion also creates an antibiotic susceptible subpopulation within the host tissues. In the absence of antibiotic therapy, the latter cells are generally more numerous than the persisters and are thought to drive chronic infections (*Claudi et al., 2014*; *Kaiser et al., 2014*; *Lawley et al., 2008*; *Monack et al., 2004*). Moreover, cells that constitute tissue reservoirs during chronic infection are known to re-seed the gut lumen over time (*Lam and Monack, 2014*). However, the role of this second, non-persister, tissue-lodged pathogen population in plasmid dissemination remained to be formally established.

Although very specific characteristics define antibiotic persistence (i.e. biphasic killing curves defining the susceptible and tolerant subpopulations) (*Balaban et al., 2019*), there are many similarities to persistent (aka long-term, chronic) infection. Numerous bacterial pathogens can survive or evade host immune defenses and antibiotics alike in a recalcitrant state (e.g. *Pseudomonas aeruginosa*, *Escherichia coli*, *Staphylococcus aureus*, or *Salmonella enterica*) (*Bakkeren et al., 2020*; *Gollan et al., 2019*; *Grant and Hung, 2013*). Using mouse models, tissue reservoirs of *S.*Tm antibiotic persisters located in the intestinal mucosa or systemic sites can re-seed the gut lumen after the cessation of antibiotic therapy (*Bakkeren et al., 2019*; *Diard et al., 2014*; *Endt et al., 2012*; *Kaiser et al., 2014*; *Onwuezobe et al., 2012*). In chronically infected hosts, the pathogen population tends to diversify forming non-growing and growing subpopulations which are kept in check by immune defenses, but can rise upon neutralization of cytokines (e.g. IFNγ or TNFα) (*Monack et al., 2004*; *Pham et al., 2020*). During monoclonal infections characteristic of many experimental models, intra-species priority effects from *S.*Tm colonizing a given site within the host reduce the chances of colonization by pathogen cells that are arriving later (*Lam and Monack, 2014*). However, gut colonization is rarely a monoclonal process. Examples of multiple Enterobacteriaceae strains co-occuring in the same host have been demonstrated in longitudinal clinical studies monitoring fecal Enterobacteriaceae populations, but also shown in swine where different strains of *Salmonella* colonize the lymph nodes compared to the gut lumen (*León-Sampedro et al., 2021*; *Martinson et al., 2019*; *San Román et al., 2018*; *Tenaillon et al., 2010*). Based on these considerations, we hypothesized that repeated re-seeding and re-invasion events during persistent infection may promote plasmid spread. Here, we define 're-seeding' as bacterial cells exiting tissue reservoirs into the gut lumen, and 're-invasion' as bacterial cells entering tissue reservoirs from the gut lumen.

Finally, we reasoned that certain resistance plasmids should themselves expedite re-seeding-transfer-invasion cycles within the infected host. In particular, this should pertain to plasmid-encoded resistance genes encoding enzymes degrading antibiotics with an extra-cytoplasmic target (e.g.

beta-lactamases). Extended spectrum beta-lactamases (ESBL) capable of hydrolyzing a wide variety of beta-lactam antibiotics including ampicillin and cephalosporins are of particular concern, as beta-lactam antibiotics account for about two-thirds of the antibiotics deployed annually worldwide (*World Health Organization, 2017*). These periplasmic enzymes may facilitate re-seeding of sensitive recipients, as resistance plasmids encoding beta-lactamases are able to reach high density under beta-lactam treatment, and clear the local environment from antibiotics, as shown in vitro (*Perlin et al., 2009*). However, it had remained unclear if such donor-mediated reduction of local beta-lactam concentrations would suffice for promoting survival of susceptible recipients in vivo, fostering plasmid transfer and subsequent reservoir formation in the host's tissue.

To address this, we quantitatively assessed the link between within-host plasmid transfer, invasion-reservoir formation-re-seeding cycles and plasmid accumulation in the host's tissues using *S*.Tm mouse models for both antibiotic persistence and chronic infection. To study these processes in a proof-of-concept approach, we chose donor-recipient-plasmid combinations featuring a high conjugation efficiency and relieved potential restrictions to conjugation or reservoir formation in the mouse gut-like colonization resistance conferred by the microbiota, and priority effects of strains that already colonize tissue reservoirs. Thereby, we could demonstrate expansion of the plasmid diversity in the gut tissue reservoir, a role for the chronically infecting *S*.Tm subpopulation and the effect of gut luminal antibiotic degradation by beta-lactamase expressing donor strains.

## Results

### Plasmids that enter *S*.Tm in the gut lumen can be stored in intestinal tissue reservoirs

To study plasmid transfer within a host and its subsequent storage in the host tissues, we modified a mouse model from our previous work (*Bakkeren et al., 2019*). For most plasmid transfer experiments, we used the IncI1 plasmid P2 (a close relative of ESBL plasmids; also known as pCol1b9), which naturally occurs in *S*.Tm SL1344 and spreads in the gut of antibiotic-pretreated mice and during diet-elicited Enterobacteriaceae blooms (*Bakkeren et al., 2021a*; *Bakkeren et al., 2019*; *Diard et al., 2017*; *Moor et al., 2017*; *Stecher et al., 2012*; *Wotzka et al., 2019*).

For all experiments, we used streptomycin-resistant derivatives of *S*.Tm ATCC 14028S that naturally do not contain P2 as recipient strains. We infected 129/SvEv mice intraperitoneally (i.p.) with recipients to allow the formation of tissue reservoirs of *S*.Tm (*Figure 1A*; left side, green) (*Bakkeren et al., 2019*). The advantage of this i.p. infection model is threefold. First, it allows us to study plasmid transfer and subsequent tissue storage in models for antibiotic persistence (the focus of *Bakkeren et al., 2019*) and to extend the work to persistent infection (this work). This is attributable to the capacity of the *Nramp1*-positive 129/SvEv mice to limit systemic pathogen growth, as shown previously in chronic infections initiated via the intravenous or the oral route (*Cunrath and Bumann, 2019*; *Monack et al., 2004*). Second, i.p. infection bypasses the need for gut colonization, ensuring that plasmid transfer observed later in the experiment is truly a result of plasmid conjugation into recipients entering from tissue reservoirs into the gut lumen. Third, the i.p. infection model mimics tissue reservoirs that are established after systemic spread following gut colonization (and therefore allows us to study the role of persistent infection in plasmid transfer dynamics) (*Bakkeren et al., 2019*; *Lam and Monack, 2014*; *Monack et al., 2004*; *Stecher et al., 2006*). Notably, our model differs from established persistent infection models in which infections are established for >28 days from an oral inoculation (*Monack et al., 2004*). However, to allow us to investigate re-seeding with high sensitivity, it was important to keep the gut lumen free of the initial recipient (without using antibiotics). We would like to point out a caveat of this i.p. model, in that it eliminates the contribution of priority effects attributable to the inoculation route. As natural infections would occur via the oral colonization route, sites that are normally occupied by invading *S*.Tm cells (e.g. the mesenteric lymph node; mLN) are less colonized after an i.p. infection compared to an oral infection. This means that invasion events from the gut luminal side later in the experiment are likely over-estimated in this model. On the other hand, this provides us with an exquisitely sensitive system to ask if tissue re-invasion can in principle expand the plasmid reservoirs within a host.

Next, we treated mice with a dose of streptomycin to suppress the microbiota and allow colonization of P2$^{cat}$ carrying donor strains (Sm$^R$) introduced orally (*Figure 1A*, blue). Note that both the

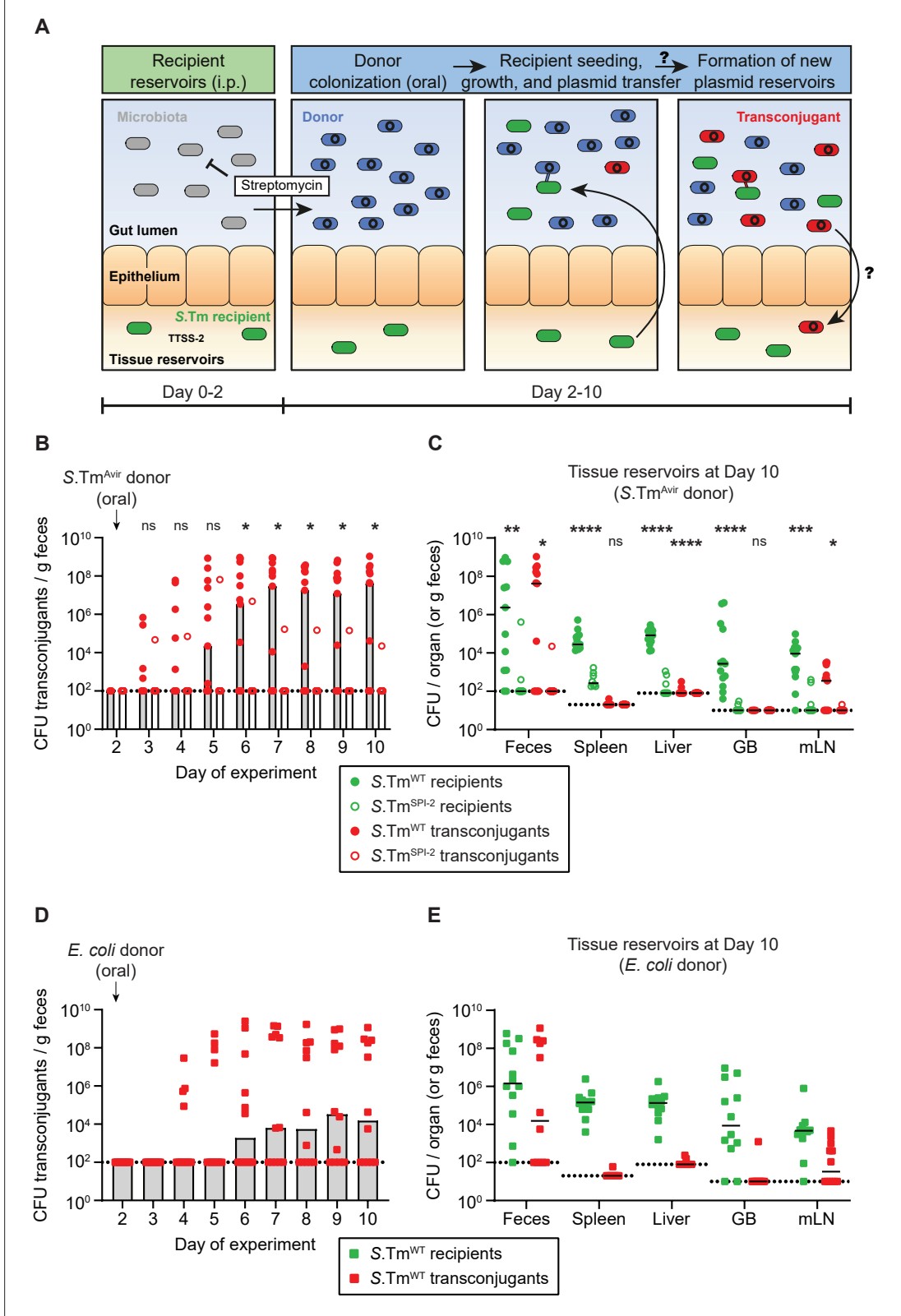

**Figure 1.** *Salmonella enterica serovar Typhimurium* (*S*.Tm) in tissue reservoirs can re-seed the gut lumen, obtain a plasmid from Enterobacteriaceae in the gut lumen, and form new plasmid-bearing reservoirs. (**A**) Proposed hypothesis. *S*.Tm recipients (green) establish tissue reservoirs after intraperitoneal injections and spread to organs, dependent on the Type three secretions system-2 (TTSS-2; encoded on SPI-2). The microbiota (grey) provides colonization resistance against colonization of the gut lumen. Donors (blue) colonize the gut lumen by an oral infection following

*Figure 1 continued on next page*

*Figure 1 continued*

a streptomycin treatment to suppress the microbiota. *S*.Tm recipients re-seed from their reservoirs and obtain a plasmid from donors forming a transconjugant (red). These transconjugants then form new tissue reservoirs. (**B**) Transconjugants are formed in the gut lumen. Mice were i.p. infected with $10^3$ CFU of a 1:1:1 mix of 14028S Sm$^R$ *TAG1-3* (*S*.Tm$^{WT}$; n = 13) or $10^3$ CFU of 14028S$^{SPI-2}$ SmR (*S*.Tm$^{SPI-2}$; *ssaV* mutant; n = 8). On day 2 post infection, 20 mg of streptomycin was given 4 hr before $5 \times 10^7$ CFU of a 1:1:1:1 mix of *S*.Tm$^{Avir}$ (*invG ssaV* mutant) P2$^{TAG4-7}$. Selective plating was used to determine the fecal transconjugant loads. (**C**) Tissue reservoirs of *S*.Tm. Organs of mice in panel B were analysed for recipients (green circles) or transconjugants (red circles). (**B–C**) Dotted line indicates the detection limit. A two-tailed Mann-Whitney U-test was used to compare *S*.Tm$^{WT}$ (solid circles) to *S*.Tm$^{SPI-2}$ (hollow circles) in each sample ($p > 0.05$ not significant (ns), $p < 0.05$ (*), $p < 0.01$ (**), $p < 0.001$ (***), $p < 0.0001$ (****)). (**D**) *S*.Tm recipients can obtain P2$^{cat}$ from *Escherichia coli*. Mice were i.p. infected with $10^3$ CFU of 14,028S Sm$^R$ *TAG1* (*S*.Tm$^{WT}$; n = 12). On day 2 post infection, 20 mg of streptomycin was given 4 hr before $5 \times 10^7$ CFU of *E. coli* 536 P2$^{cat}$. Selective plating and colony colour on MacConkey agar was used to determine the fecal transconjugant loads. (**E**) Tissue reservoirs of *S*.Tm after conjugation with *E. coli*. Organs in panel B were analysed for recipients (green squares) or transconjugants (red squares). (**D–E**) Dotted line indicates the detection limit. (**B,D**) Bars indicate median. (**C,E**) Lines indicate median. GB = gall bladder; mLN = mesenteric lymph node. (**B–E**) Fecal populations of donors, recipients, and transconjugants are presented in *Figure 1—figure supplement 1*.

The online version of this article includes the following source data and figure supplement(s) for figure 1:

**Source data 1.** CFU data for *Figure 1*.

**Figure supplement 1.** Fecal populations for mice in *Figure 1*.

**Figure supplement 1—source data 1.** CFU data for *Figure 1—figure supplement 1*.

**Figure supplement 2.** Verification of transconjugant formation by colony PCR.

**Figure supplement 3.** Type three secretions system-1 (TTSS-1)-dependent invasion of recipients is not necessary for re-seeding.

**Figure supplement 3—source data 1.** CFU data for *Figure 1—figure supplement 3*.

**Figure supplement 4.** Donor inoculation does not promote re-seeding.

**Figure supplement 4—source data 1.** CFU data for *Figure 1—figure supplement 4*.

**Figure supplement 5.** Control for re-seeding and obtaining a plasmid from persisters induced by antibiotic treatment.

**Figure supplement 5—source data 1.** CFU data for *Figure 1—figure supplement 5*.

**Figure supplement 6.** Localization of tissue reservoirs of *Salmonella enterica* serovar Typhimurium (*S*.Tm) prior to re-seeding.

**Figure supplement 6—source data 1.** CFU data for *Figure 1—figure supplement 6*.

donor and recipient strains used in this experimental model are streptomycin-resistant. Therefore, the systemic infection by the recipients is unaffected by the antibiotic and the recipients can grow upon re-seeding of the gut lumen (=persistent systemic infection model). As the donor, we used a derivative of *S*.Tm SL1344 that lacks functional type three secretions systems (TTSS)-1 and -2 (*invG* and *ssaV* mutant; SL1344$^{Avir}$) (*Hapfelmeier et al., 2005*). This prevented invasion of the *S*.Tm donors into tissues to exclude tissue-localized plasmid transfer. We used SL1344 derivatives to study interactions between different strains of bacteria, rather than isogenic strain interactions that would limit co-existence of donors and recipients in the gut after re-seeding (*Lam and Monack, 2014*; *Lee et al., 2013*). Feces were monitored over time to investigate plasmid transfer dynamics and mice were euthanized at day 10 post infection to assess the formation of new intracellular plasmid reservoirs by recipients that had obtained a plasmid (transconjugants; *Figure 1A*, red).

One day after the addition of donors (SL1344$^{Avir}$ P2$^{cat}$), recipients started re-seeding into the gut lumen and transconjugants were detected in the feces (*Figure 1B*, *Figure 1—figure supplement 1A*). By day 10 of the experiment, transconjugants were detected in the majority of mice (8/13 mice; *Figure 1B*). We confirmed that plasmid transfer had occurred by performing PCRs with both plasmid- and recipient-specific primers (*Figure 1—figure supplement 2*). Re-seeding and plasmid acquisition depended on the TTSS-2 encoded on *Salmonella* pathogenicity island (SPI)-2, as recipients lacking a functional TTSS-2 (14028S *ssaV* mutant; *S*.Tm$^{SPI-2}$) failed to re-seed frequently, and consequently transconjugants were rarely observed (1/8 mice; *Figure 1*; *Figure 1—figure supplement 1B*). Their failure to establish substantial tissue reservoir sizes (e.g. in the spleen, liver, or gall bladder) could explain the limited re-seeding of *S*.Tm$^{SPI-2}$ (*Figure 1C*). However, re-seeding was not dependent on the TTSS-1 encoded on SPI-1, as mutant recipient strains that lack a functional TTSS-1 apparatus (14028S *invG* mutant; *S*.Tm$^{SPI-1}$) established tissue reservoirs and re-seeded the gut (*Figure 1—figure supplement 3*). Furthermore, we performed a control experiment to exclude the contribution of donor strains themselves to the re-seeding process of recipients. To this end, we performed the same experiment as in *Figure 1A*, but in parallel included a control group where donors were not added. Re-seeding was detected in both groups (2/4 mice with donors; 3/5 mice without donors by day 10 of

the experiment), indicating that the presence of donors did not promote re-seeding (*Figure 1—figure supplement 4*).

Altogether, these results indicated that bacteria re-seeding from tissue reservoirs can gain plasmids in the gut. In contrast to previous work (*Bakkeren et al., 2019*), this occurred without previous selection for persister subpopulations. Performing our experiments in the absence of antibiotic treatment to select for persisters allowed us to assess the contribution of growing tissue-lodged pathogen subpopulations. Importantly, we analysed here if the gut luminal plasmid was carried back into host tissues by transconjugants. Indeed, transconjugants were detected in the mLNs of mice where gut luminal plasmid transfer was observed. Occasionally such transconjugants were also observed in other tissue reservoirs such as the spleen and liver associated with further systemic spread after tissue invasion (*Figure 1C*).

Next, we tested if this process could be generalized to other donors colonizing the gut lumen. For this, we used *E. coli* 536 carrying P2$^{cat}$ as a donor. This strain is naturally streptomycin-resistant and colonizes the streptomycin-pretreated mouse gut for long periods of time (*Brzuszkiewicz et al., 2006*; *Ghalayini et al., 2019*). Also in this case, re-seeding of *S*.Tm 14028S recipients occurred followed by the formation of transconjugants in the majority of the mice (7/12 mice) (*Figure 1D*, *Figure 1—figure supplement 1C*). Once again, re-seeding and conjugation was associated with the formation of new plasmid tissue reservoirs by invading transconjugants, primarily in the mLN (*Figure 1E*). To assess if our i.p. infection experimental model allowed the comparison between both persistent infection and antibiotic persistence, we performed a control experiment in which we treated mice three times with ceftriaxone 2 days after i.p. infection to select for persisters (*Figure 1—figure supplement 5*), as done previously (*Bakkeren et al., 2019*). After the addition of *E. coli* 536 P2$^{cat}$ donors, *S*.Tm$^{WT}$ recipients began to re-seed in some mice. However, this occurred later and in a smaller number of the analysed mice, as antibiotic persisters are much less abundant (and need to re-enter growth) compared to the much larger population of persistently infecting *S*.Tm$^{WT}$ recipient cells. Regardless, this re-seeding was also associated with plasmid transfer and the occasional formation of tissue reservoirs of transconjugants in the mLN (*Figure 1—figure supplement 5*).

Altogether, our data demonstrates that plasmids can subvert *S*.Tm derived both from tissue-lodged populations of antibiotic persisters and from persistent infections. Cells from either type of tissue-lodged population can re-enter the gut lumen and serve as recipients in interbacterial interactions. This process can be followed by re-invasion forming new tissue reservoirs, capturing a record of conjugative plasmids that had previously been present in the gut lumen.

## Re-seeding of the gut lumen, plasmid transfer, and the formation of new reservoirs is limited by the gut luminal carrying capacity and the conjugation rate

Next, we examined the dynamics of pathogen-assisted plasmid transfer and storage. This should identify bottlenecks and inform about the possible routes that recipients and transconjugants take to exit and re-enter host tissues in more detail. First, we explored possible sources of persistently infecting recipients that re-seed the gut lumen by characterizing the localization of *S*.Tm in tissue reservoirs. We infected mice i.p. with *S*.Tm 14028S recipients and euthanized mice 2 days later, while keeping the gut luminal microbiota intact (at the time point when normally the donors are added). As expected based on previous work (*Bakkeren et al., 2019*; *Grant et al., 2008*; *Hensel et al., 1995*; *Lam and Monack, 2014*; *Monack et al., 2004*), *S*.Tm were found in the spleen, liver, and kidney at high density (*Figure 1—figure supplement 6*), which likely occurred because of blood flow to these organs immediately after i.p. infection of *S*.Tm (*Grant et al., 2008*). Although bacteria were not detected in the feces (our readout for re-seeding), *S*.Tm was found in the content of the small intestine (i.e. primarily the jejunum and ileum, but also the duodenum), and this was correlated with *S*.Tm in the tissues of these sites (*Figure 1—figure supplement 6*). Furthermore, we found high densities of *S*.Tm in the lining of the small intestinal tissues that contains the gut-associated tissues such as Peyer's patches (here called small intestinal gut-associated lymphoid tissues [S.I. GALT]). The gall bladder contained some *S*.Tm, but the low densities suggest that the gall bladder is not the primary route of re-seeding in this model (as suggested in some other experimental models *Lam and Monack, 2014*). Future work may address if *S*.Tm re-seeds from the small intestinal tissues after spread from blood or lymphatic supply.

Instead, we focused on the dynamics of re-seeding followed by plasmid transfer into this luminal pathogen population and asked in which conditions this occurs. We mixed equal ratios of recipient strains harbouring three different sequence-tagged barcodes at a neutral location in the chromosome (14028S Sm$^R$ TAG1-3; 1:1:1 ratio) and i.p. infected mice (same mice as in *Figure 1B*). As donor strains, we used SL1344$^{Avir}$ containing P2 labelled with four additional unique sequence tags (P2$^{TAG4-7}$; 1:1:1:1 ratio). All tags can be identified using quantitative PCR (qPCR) (*Grant et al., 2008*). At the end of the experiment, we analysed the ratio of each of these tags in the recipient population (enriching for recipients = recipients + transconjugants) by qPCR. We plotted the relative proportion of each tag relative to all tags (both chromosomal and recipient tags) to preserve information about the proportion of plasmid tags within the recipient population. To visually correlate re-seeding of *S*.Tm into the gut lumen with the localization of these tags in tissue reservoirs, we sorted and color-coded the tags according to the most abundant tag in the feces. While chromosomal tags were evenly distributed in primary tissue reservoirs (e.g. spleen and liver), the feces were typically dominated by just one of the three tagged strains, implying a narrow population bottleneck (*Figure 2A*, *Figure 2—figure supplement 1*). Thus, re-seeding from these tissue reservoirs is a rare process that is followed by clonal expansion. We also analysed the gall bladder as a possible source of re-seeding, and the mLN as a site where new tissue reservoirs are formed after re-invasion of re-seeding *S*.Tm. Although the chromosomal tags were slightly more skewed in the gall bladder compared to the spleen and liver, there was no correlation between the rank of the tags found in the feces and in the gall bladder (*Figure 2A*, *Figure 2—figure supplement 1A*; e.g. mouse 7 contains different most abundant tags in the feces and gall bladder). Interestingly, the mLN and the feces typically shared the same most abundant chromosomal tag (~10-fold median reduction of the proportion of the most and second most abundant tag in the feces; *Figure 2A*), suggesting that re-seeding followed by re-invasion could be skewing this distribution, rather than a bottleneck in cells spreading to the mLN after the i.p. injection. Note that the mLN population likely reflects the population that has invaded into the gut tissue. However, due to unavoidable background contamination from separating the gut tissue from the gut lumen, we chose to examine the mLN instead. The correlation between mLN and fecal chromosomal tags was not seen in one mouse (mouse 9 in *Figure 2—figure supplement 1A*), in which recipients were not found at high densities in the feces, further supporting the hypothesis that the distribution of tags in the mLN is heavily influenced by re-invading cells from the gut lumen. We plotted the plasmid tags found in the transconjugant population in the same manner (*Figure 2B*). In mice where plasmid transfer was detected (population sizes in *Figure 1—figure supplement 1A*; raw tag data in *Figure 2—figure supplement 1*), almost all plasmid tags were found, although their distribution was skewed (*Figure 2B*). Importantly, the plasmid tags found in the feces were mostly also found in the mLN, implying that plasmids were carried into the mLN by transconjugants that had formed in the gut lumen (*Figure 2B*). In addition, the data indicated that mLN entry of the plasmid-bearing transconjugants occurred more than once in most of the mice.

As a control, we also analysed the extent to which the presence of donors in the gut lumen contributed to the rarity of re-seeding, analyzing the seven chromosomal tags contained in the recipient population of mice in *Figure 1—figure supplement 4*. We found that one or few tags dominated the population in the feces in both mice with or without donors, indicating that the presence of donors did not have a detectable effect on the low re-seeding frequency (*Figure 2—figure supplement 2*).

Next, we developed a mathematical model (Supplementary information) to explore the relative contribution of re-seeding (migration rate; μ) and plasmid transfer (conjugation rate per mating pair; donor or transconjugant to recipient; γ) to the bacterial dynamics. We fit the model to the experimental data (evenness of tags in *Figure 2A–B*; recipient and transconjugant population sizes in *Figure 1—figure supplement 1A*; Supplementary information) to reveal the most likely migration and conjugation rates (*Figure 2C* in red; μ = 1.78 recipient CFU/g feces per day; γ = 3.16 × 10$^{-11}$ events per CFU/g feces per day; *Appendix 1—table 2*). This corresponds to an average per-recipient rate of migration vs. conjugation (μ, γ) of 2.1 × 10$^{-6}$ per day (by dividing μ with the recipient population in host tissue reservoirs) or 3.2 × 10$^{-2}$ per day respectively (by multiplying γ with the gut luminal donor population; Supplementary information). This identified recipient re-seeding as the rate limiting step in our observed plasmid transfer dynamics, although the conjugation rate influenced the evenness of the plasmid tags and the time until recipients were converted into transconjugants (*Figure 2—figure supplement 3*). In our experimental system, we used streptomycin to decrease colonization

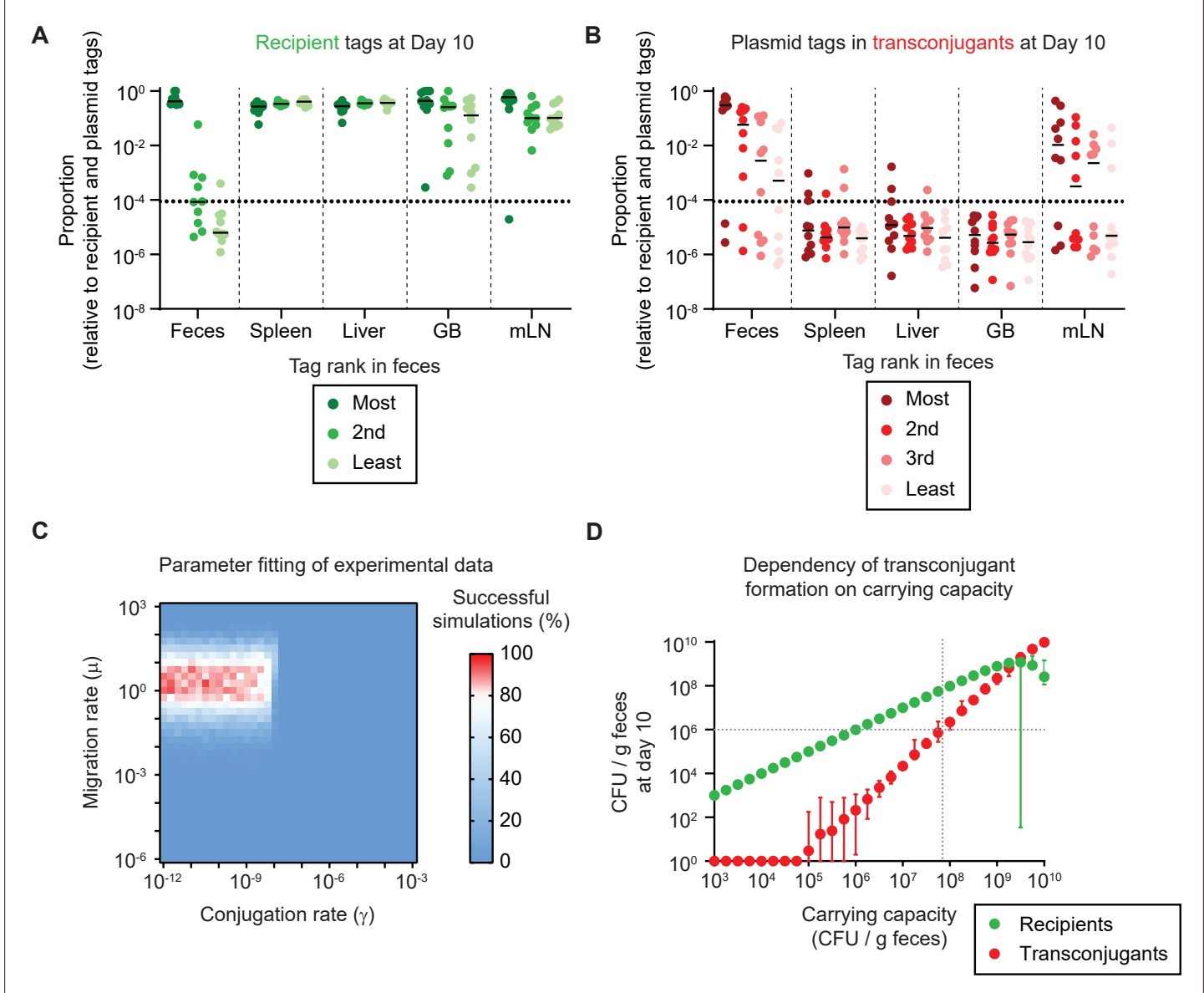

**Figure 2.** Re-seeding from tissue reservoirs is a rate limiting step that suffices in conjugation and formation of new reservoirs given a sufficient carrying capacity. (**A–B**) Mice in *Figure 1* where re-seeding of recipients occurred (n = 10) were analysed for the distribution of sequence tags in enrichments of recipients at day 10 of the experiment (including transconjugants) by quantitative PCR (qPCR) (raw tag data shown in *Figure 2—figure supplement 1*). The dotted line indicates the conservative detection limit. Lines indicate the median. The proportions of tags are plotted relative to all seven tags (three recipient and four plasmid tags). (**A**) Recipient tags and (**B**) plasmid tags were sorted according the most abundant recipient or plasmid tag in the feces, in each mouse. GB = gall bladder; mLN = mesenteric lymph node. (**C**) Fitting of simulations to experimental data. Individual summary statistics used to fit simulations are shown in *Figure 2—figure supplement 3*. Parameters and method for fitting is described in Supplementary information. The migration rate of μ = 1.78 recipient CFU/g feces per day and the conjugation rate of γ = 3.16 × 10⁻¹¹ events per CFU/g feces per day provide the best fit to the experimental data (red indicates rate pairs that fit the experimental data most often). (**D**) The migration and conjugation rates were fixed to the most likely values in panel C and the carrying capacity was varied. Recipient (green) and transconjugant (red) populations at the end of the simulation (day 10; n = 100 simulations per carrying capacity) are shown as the mean with the range of values. The lowest value was set to 1 CFU/g feces to allow visualization on a log scale. The grey dotted lines correspond to the carrying capacity at which transconjugants are present at 10⁶ CFU/g feces, as a density threshold that we speculate should facilitate some frequency of invasion into tissue reservoirs.

The online version of this article includes the following source data and figure supplement(s) for figure 2:

**Source data 1.** Tag frequency data for *Figure 2*.

**Figure supplement 1.** Raw tag proportions before sorting and re-colouring for mice in *Figure 2*.

**Figure supplement 1—source data 1.** Raw tag frequency data for *Figure 2—figure supplement 1*.

*Figure 2 continued on next page*

*Figure 2 continued*

**Figure supplement 2.** Distribution of tag proportions dependent on the presence or absence of donors.

**Figure supplement 2—source data 1.** Raw tag frequency data for *Figure 2—figure supplement 2*.

**Figure supplement 3.** Summary statistics used for fitting the mathematical model to the experimental data in *Figure 2C*.

**Figure supplement 4.** Dependence of recipient and transconjugant population sizes on the carrying capacity with a higher or lower conjugation rate compared to *Figure 2D*, and correlation with experimental data in *Figure 1*.

**Figure supplement 4—source data 1.** Data used for correlations in *Figure 2—figure supplement 4*.

resistance, effectively raising the carrying capacity (parametrized at $10^9$ CFU/g feces for *Figure 2C*, *Figure 2—figure supplement 3*). However, since conjugation is density-dependent, we reasoned that the level of colonization resistance should influence our results. To test the range of Enterobacteriaceae densities that would allow appreciable densities of transconjugants (e.g. sufficient to allow invasion into new tissue reservoirs), we fixed the migration and conjugation rates to the most likely values from *Figure 2C* (i.e. $\mu = 1.78$ recipient CFU/g feces per day; $\gamma = 3.16 \times 10^{-11}$ events per CFU/g feces per day). We then varied the carrying capacity as a proxy for colonization resistance, using the population sizes of recipients and transconjugants at the end of the experiment (day 10) as a readout. While recipient densities increased proportionally to the carrying capacity, the transconjugant population size remained below $10^6$ CFU/g feces until the carrying capacity reached $10^7$–$10^8$ CFU/g feces (*Figure 2D*; grey dotted lines). To validate this with our experimental data, we re-analysed the experimental data from *Figure 1B–E* (pooling data from both *S*.Tm$^{Avir}$ and *E. coli* donors), correlating the donor population size at day 10 of the experiment (as a proxy for colonization resistance) with the transconjugant populations in both the feces and the mLN at day 10 (*Figure 2—figure supplement 4A-B*). As suggested by our model, transconjugant formation and re-invasion was more likely in mice that harboured higher gut luminal donor densities, indicating a reduction in colonization resistance (*Figure 2—figure supplement 4A-B*). We confirmed this trend in simulations with both a higher and lower conjugation rate than in our experimental system (*Figure 2—figure supplement 4C-D*). A plasmid with a higher conjugation rate was found to be less inhibited by colonization resistance. It could rise to appreciable densities (e.g. >$10^6$ CFU/g) even with lower overall carrying capacities. Altogether, our data suggest a critical role for colonization resistance in limiting tissue-reservoir-assisted plasmid spread dynamics, but also highlights the role of the plasmid transfer rate defined by the plasmid-strain association.

### Plasmids can use re-invading *S*.Tm to form a persistent tissue reservoir that survives antibiotics and permits spread to further strains in the gut post-antibiotic treatment

We then addressed the implications of newly formed plasmid tissue reservoirs for future plasmid transfer dynamics in the gut. For this, in the first phase of the experiment, we performed the same experiment as in *Figure 1D* (using *S*.Tm 14028S Sm$^R$ as a recipient and *E. coli* 536 P2$^{cat}$ as a donor), but on day 10 we gave mice three doses of ciprofloxacin instead of stopping the experiment (second phase of the experiment; *Figure 3A*). As before, *S*.Tm re-seeded from tissue reservoirs, obtained a plasmid in the gut lumen, and transconjugants carried the plasmid into host tissues (i.e. mLN) before the ciprofloxacin treatment killed gut luminal populations and left only tissue-associated persisters (including some of the newly formed *S*.Tm 14028S Sm$^R$ P2$^{cat}$ cells) to survive (*Figure 3A*, *Figure 3—figure supplement 1A*). We used ampicillin in the drinking water to suppress re-seeding of these tissue-associated reservoirs before the third phase of the experiment. In this third phase, we introduced a secondary recipient (SL1344 $\Delta$P2) and monitored plasmid transfer (and thus the formation of secondary transconjugants) to this strain. Strikingly, secondary transconjugants were formed in most mice (3/5 mice, numbered 1–3; *Figure 3A*, *Figure 3—figure supplement 1A*; black dots), correlating with both the presence of primary transconjugants in newly formed tissue reservoirs in the mLN and re-seeding of primary transconjugants (*Figure 3B*; *Figure 3—figure supplement 1B*). Importantly, *E. coli* donors were not detected in this experiment after the ciprofloxacin treatment, suggesting that plasmid transfer was due to re-seeding primary transconjugants from their new reservoirs (*Figure 3—figure supplement 1A-B*). We performed a series of control experiments to support that plasmid transfer to secondary recipients was due to the re-seeding of primary transconjugants. In one control

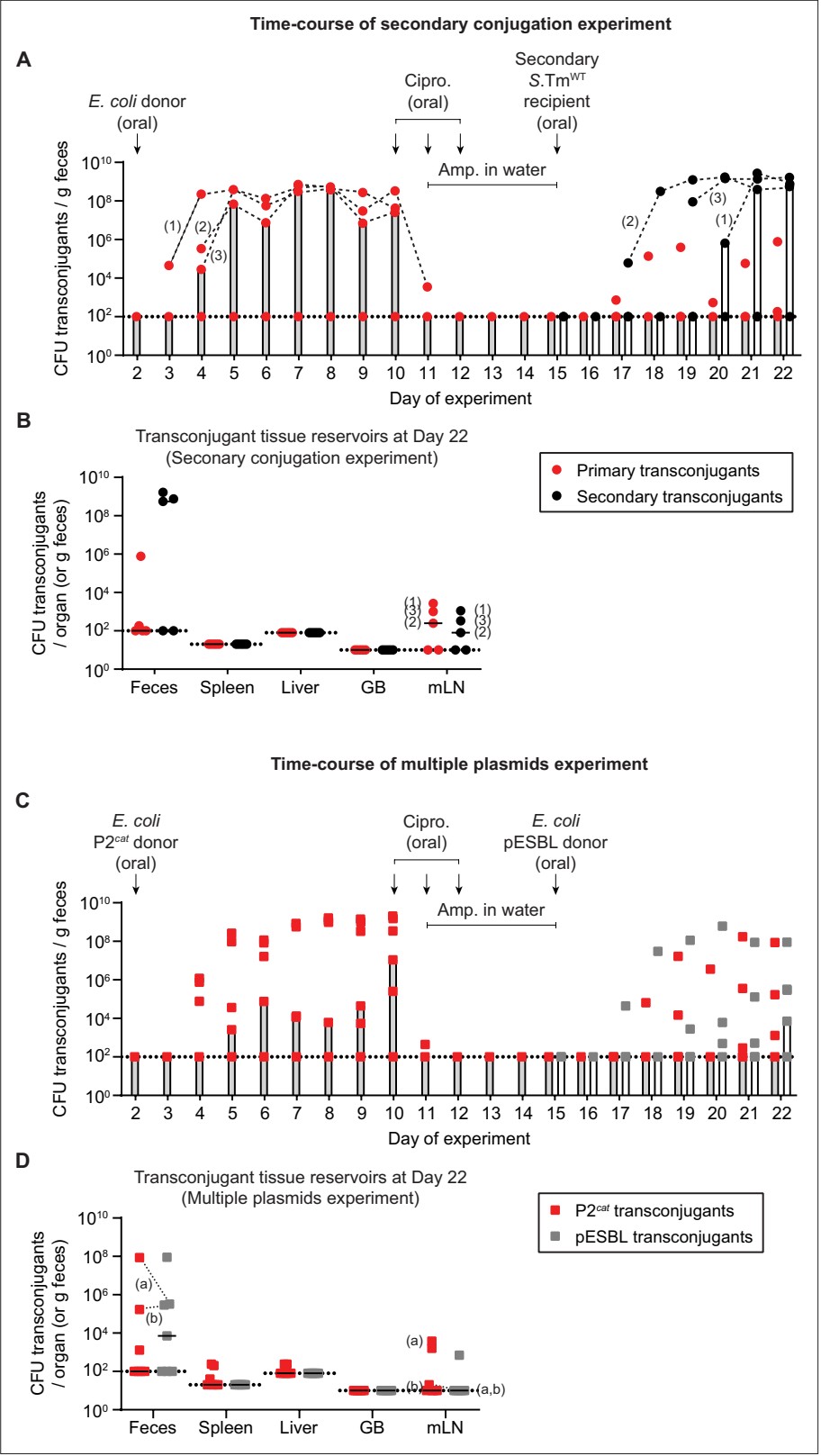

**Figure 3.** Newly formed plasmid reservoirs can spread plasmids to recipients after antibiotic treatment and plasmids can accumulate in the gut dependent on tissue reservoirs. (**A**) *Salmonella enterica* serovar Typhimurium (*S*.Tm) transconjugants can conjugate to secondary recipients after survival of antibiotic treatment, by survival in tissue reservoirs. Mice were i.p. infected with $10^3$ CFU of 14028S Sm$^R$ *TAG1* (Kan$^R$; n = 5). On day 2 post infection,

*Figure 3 continued on next page*

*Figure 3 continued*

20 mg of streptomycin was given 4 hr before $5 \times 10^7$ CFU of *Escherichia coli* 536 P2$^{cat}$ (CmR). Selective plating and colony colour on MacConkey agar was used to determine the fecal transconjugant loads (red circles). On day 10, mice were given 3 mg ciprofloxacin for 3 days in a row, along with 2 g/l ampicillin in the drinking water from day 11 to 15. On day 15, a secondary recipient (SL1344 ΔP2; Amp$^R$) was added ($5 \times 10^7$ CFU oral). Selective plating allowed the enumeration of secondary transconjugants (black circles). Bars indicate the median. Dashed black lines connect primary or secondary transconjugant populations of the same mice (mice are numbered 1–3; the same numbering is used in panel B). (**B**) Tissue reservoirs of *S*.Tm. Organs of mice in panel A were analysed for primary transconjugants and secondary transconjugants. Lines indicate the median. GB = gall bladder; mLN = mesenteric lymph node. The numbers correspond to the same mouse numbering as in panel A. (**C**) *S*.Tm recipients can allow the formation of multiple transconjugants over time dependent on multiple re-seeding events. Mice (n = 7) were i.p. infected with recipients 14028S Sm$^R$ *TAG1* (Kan$^R$) and donors *E. coli* 536 P2$^{cat}$ (CmR) as in panel A. On day 10, mice were given antibiotics as in panel A. On day 15, a second donor, (*E. coli* pESBL; Amp$^R$) was added ($5 \times 10^7$ CFU oral). Selective plating allowed the enumeration of both P2$^{cat}$ (red squares) and pESBL transconjugants (grey squares). Bars indicate the median. (**D**) Tissue reservoirs of *S*.Tm. Organs of mice in panel C were analysed for P2$^{cat}$ and pESBL transconjugants. Lines indicate the median. GB = gall bladder; mLN = mesenteric lymph node. Dashed lines connect the two transconjugant populations in the same fecal sample (two mice harbor both types of transconjugants: mouse (**a**) and (**b**); these mouse labels are also shown on the mLN population). (**A–D**) Dotted lines indicate the detection limits. Population sizes of all strains and subpopulations are presented in *Figure 3—figure supplement 1*.

The online version of this article includes the following source data and figure supplement(s) for figure 3:

**Source data 1.** CFU data for *Figure 3*.

**Figure supplement 1.** Additional fecal and organ populations for mice in *Figure 3*.

**Figure supplement 1—source data 1.** CFU data for *Figure 3—figure supplement 1*.

**Figure supplement 2.** Control that *Escherichia coli* donors are not detected in tissue reservoirs after antibiotic treatment.

**Figure supplement 2—source data 1.** CFU data for *Figure 3—figure supplement 2*.

**Figure supplement 3.** Further control experiment to determine that *Escherichia coli 536* cannot form persister reservoirs in host tissues.

**Figure supplement 3—source data 1.** CFU data for *Figure 3—figure supplement 3*.

**Figure supplement 4.** Conjugation is not detectable within host tissue reservoirs.

**Figure supplement 4—source data 1.** CFU data for *Figure 3—figure supplement 4*.

experiment, we performed the same experiment as in *Figure 3A–B* but modified the experiment in two ways. First, we euthanized a subset of mice at day 15 (when we normally added the secondary recipient) and determined that *E. coli* donors were not present in tissue reservoirs (*Figure 3—figure supplement 2*). Second, we tested to see if *E. coli* donors could survive antibiotic treatment generally in the absence of *S*.Tm. We orally introduced donors in the absence of *S*.Tm and performed the same antibiotic treatment regime as *Figure 3A–B*. After the treatment regime, we added streptomycin to the drinking water to select for any remaining *E. coli*. No *E. coli* could be detected, neither in the feces nor in tissue reservoirs (*Figure 3—figure supplement 3*). This confirmed that it is unlikely that *E. coli* donors contributed to plasmid transfer after antibiotic treatment in *Figure 3A–B*. Lastly, we performed a different control experiment to investigate if plasmid transfer could in principle proceed within tissues. This would address if invading secondary recipients could be receiving plasmids (within host tissues) dependent on invasion into host tissues. We infected mice i.p. with both virulent *S*.Tm donors (SL1344 P2$^{cat}$) and 5 min later with recipients (14028S). This sequential inoculation ruled out conjugation in the inoculum. After 3 days, we examined the organs to quantify plasmid transfer. While both donors and recipients were detected in the analysed organs, transconjugants were not detected, correlating with the absence of gut luminal growth (*Figure 3—figure supplement 4*). Altogether, these data verified that primary transconjugants re-seeding into the gut lumen were responsible for plasmid transfer to secondary recipients in *Figure 3A*, and that plasmid transfer preceded re-invasion into host tissues.

Next, we tested if multiple plasmids could accumulate in the gut lumen from tissue reservoirs. We performed the first two phases of the experiment as above (*Figure 3A–B*), but this time in the third phase of the experiment, we added a second *E. coli* donor containing an ESBL resistance plasmid

(pESBL) shown to conjugate efficiently both in antibiotic pre-treated mice and in mice containing a reduced complexity microbiota (*Bakkeren et al., 2019*; *Benz et al., 2021*). As in the previous experiment, P2$^{cat}$ transconjugants were formed in the first phase of the experiment, followed by the elimination of bacteria from the gut lumen and storage of P2$^{cat}$ transconjugants in tissue reservoirs in the second phase of the experiment (*Figure 3C*; *Figure 3—figure supplement 1C-D*). Following the introduction of pESBL donors, pESBL transconjugants were detected in most mice (4/7 mice) as additional recipients re-seeded from their reservoirs (*Figure 3C*; *Figure 3—figure supplement 1C*). Importantly, in two of these four mice (labelled 'a' and 'b'), P2$^{cat}$ transconjugants were also detected in the feces, suggesting that P2$^{cat}$ transconjugants re-seeded from their reservoirs (*Figure 3C*; *Figure 3—figure supplement 1C*). As in the other experiments, pESBL transconjugants could also be detected in the mLN of one mouse (*Figure 3D*). These data indicate that invasive enteropathogens like *S*.Tm might allow different resistance plasmids to accumulate in tissue reservoirs in the long run. However, it is important to reiterate that invasion into the mLN is likely influenced by priority effects, making each subsequent invasion less likely. This may explain why both types of transconjugants were not found in the same mLN in the tested mice (mouse 'a' and 'b'; *Figure 3D*). Furthermore, while in this experiment both types of plasmids did not accumulate within the same bacterial cell (likely because these are both *Inc*I1 plasmids and are therefore incompatible), it is plausible that transconjugants containing multiple compatible plasmids could arise. Nevertheless, we show that the accumulation of two plasmids in the gut lumen can be facilitated by tissue reservoirs.

Altogether, our data indicate that tissue reservoirs of *S*.Tm can influence plasmid spread in the gut both by spreading plasmids to different recipients and by accumulating multiple plasmids over time.

## ESBL resistance plasmids can facilitate re-seeding of susceptible *S.* Tm recipients from tissue reservoirs under beta-lactam treatment, and thereby promote resistance plasmid spread

Our data suggest that some plasmids are selfish genetic elements that can use persistent pathogen cells to form long-term reservoirs in host tissues. We reasoned that a subset of these plasmids (e.g. those encoding for resistances that can clear antibiotics nearby, such as beta-lactamases) may have another way to promote such transconjugant-dependent reservoir formation by protecting potential recipients in their surroundings. The survival of *S*.Tm in the presence of otherwise lethal concentrations of beta-lactam antibiotics mediated by ampicillin-resistant *E. coli* has been described in vitro, based on the extracellular acting mechanism of beta-lactamases (*Perlin et al., 2009*). Furthermore, antibiotic treatment is commonly used in the treatment of bacterial infections in humans and livestock and beta-lactam antibiotics account for about two-thirds of the total antibiotic usage worldwide (*Cully, 2014*; *World Health Organization, 2014*; *World Health Organization, 2017*). Therefore, we asked if gut luminal Enterobacteriaceae harbouring plasmids that confer resistance to the antibiotic used could facilitate survival of *S*.Tm recipients long enough to obtain a plasmid and survive in the gut lumen. As a class of multi-drug resistance plasmids that has drawn particular attention as a contributor to antibiotic resistance spread, and because of their extracellular mechanism of action, we chose to study ESBL plasmids under beta-lactam treatment (*Bajaj et al., 2016*; *Carattoli, 2009*; *Davies and Davies, 2010*; *World Health Organization, 2017*).

As previously, we infected *S*.Tm recipients (14028S Sm$^R$) i.p. into mice. One day post infection, we introduced *E. coli* orally either bearing pESBL (same strains used in *Figure 3C–D*; clinical isolate characterized and sequenced in *Bakkeren et al., 2019*; *Benz et al., 2021*), or P2$^{cat}$ as a control (isogenic *E. coli* strains; the P2$^{cat}$ *E. coli* was cured of pESBL, followed by in vitro conjugation of P2$^{cat}$), and simultaneously gave the mice ampicillin in the drinking water (*Figure 4*). Treating mice with 0.5 g/l ampicillin in the drinking water corresponds to a daily dose of 2–6 g in a 70 kg human, close to the recommended treatment schedule of oral ampicillin of 500 mg every 6 hr (*Wirfs, 2019*). As expected, *E. coli* pESBL grew to high densities, whereas in the control animals inoculated with ampicillin-susceptible *E. coli* P2$^{cat}$ (negative control donors; chloramphenicol resistance), the *E. coli* could not survive (*Figure 4—figure supplement 1A*). Over time, *S*.Tm began re-seeding from their tissue reservoirs in both groups of mice (*Figure 4—figure supplement 1B*), but could only bloom in the gut lumen in the presence of pESBL *E. coli* (*Figure 4A*). This bloom was associated with transfer of the pESBL, followed by clonal expansion as *S*.Tm became resistant to ampicillin, leading to >99% of the *S*.Tm gut luminal population obtaining a plasmid (*Figure 4B*). Importantly, since *S*.Tm is an enteric

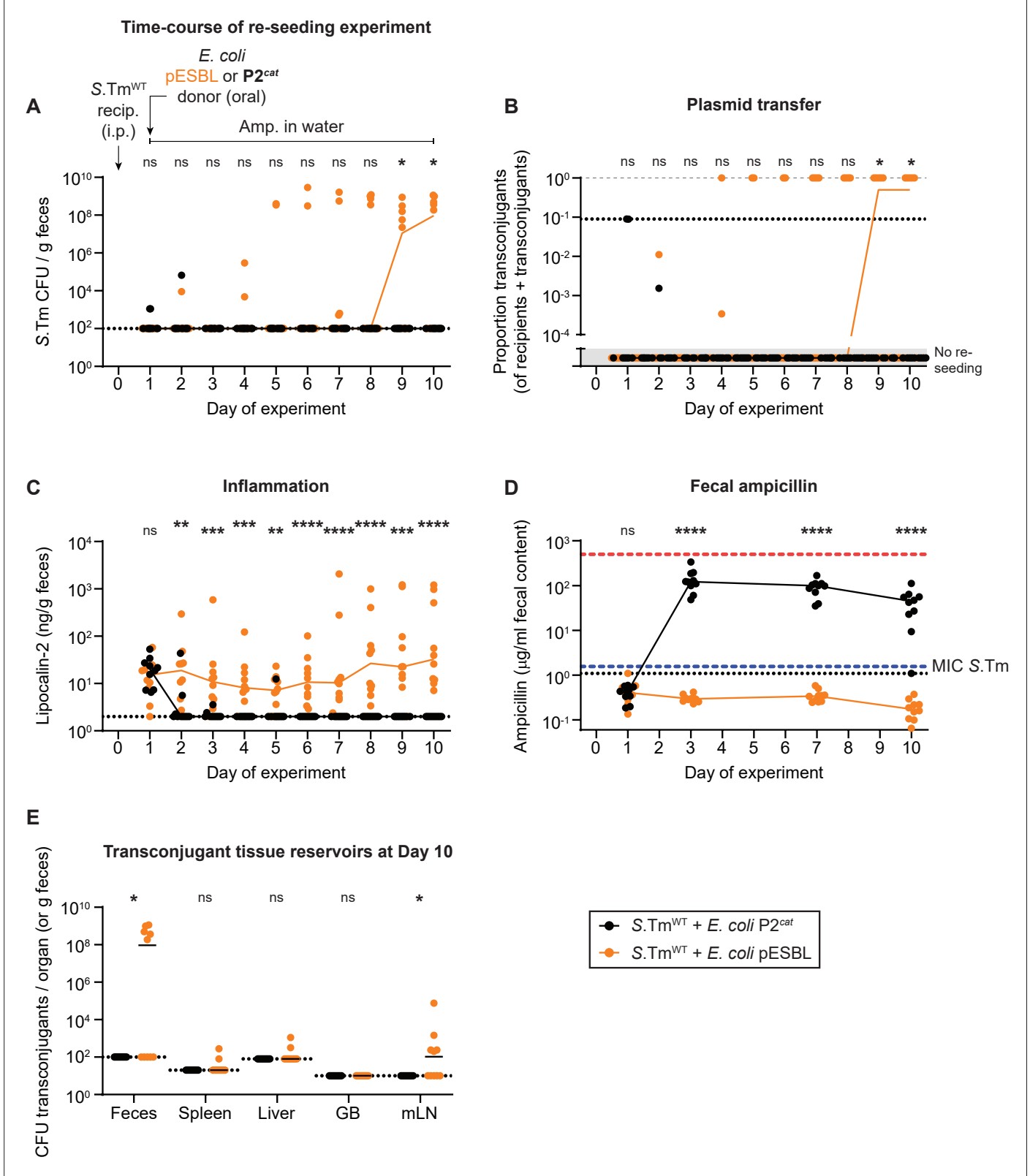

**Figure 4.** Re-seeding of *Salmonella enterica* serovar Typhimurium (*S*.Tm) followed by reception of a plasmid and storage in new tissue reservoirs can occur under beta-lactam counterselection, dependent on *Escherichia coli* conjugative extended spectrum beta-lactamases (ESBL) plasmids. (**A–E**) Mice were i.p. infected with $10^3$ CFU of a 1:1:1:1:1:1:1 mix of 14028S Sm$^R$ *TAG1-7* (Kan$^R$). On day 1 post infection, mice were given 0.5 g/l ampicillin in the drinking water and either $5 \times 10^7$ CFU of *E. coli* P2$^{cat}$ (Cm$^R$; black circles; n = 10) or *E. coli* pESBL (Amp$^R$; orange circles; n = 10). Dotted lines indicate

*Figure 4 continued on next page*

*Figure 4 continued*

detection limits. Lines connect medians on each day. A two-tailed Mann-Whitney U-test was used to compare $S.\text{Tm}^{WT}$+ *E. coli. coli* P2$^{cat}$ to $S.\text{Tm}^{WT}$+ *E. coli. coli* P2$^{cat}$ in each sample (p > 0.05 not significant (ns), p < 0.05 (*), p < 0.01 (**), p < 0.001 (***), p < 0.0001 (****)). (**A**) Re-seeding *S*.Tm was analysed by selective plating on kanamycin-containing MacConkey agar. (**B**) Conjugation was determined by selective and/or replica plating. The proportion transconjugants is calculated by the population size of transconjugants divided by the sum of both transconjugants and recipients. The grey dashed line indicates a proportion of 1. The dotted line indicates the conservative detection limit for transconjugants (since the proportion depends on the recipient population size), each sample has a different detection limit and therefore values can appear below the detection limit. Values in the grey box in the lower part of the y-axis are plotted to indicate mice with no re-seeding. (**C**) Inflammation was quantified using an ELISA for fecal lipocalin-2. (**D**) Fecal ampicillin was measured using mass spectrometry. The blue dashed line indicates the minimum inhibitory concentration of *S*.Tm in vitro (*Bakkeren et al., 2019*). The red dashed line indicates the concentration of ampicillin given to the mice in the drinking water. (**E**) Organs were analysed for transconjugant populations on day 10 of the experiment. GB = gall bladder; mLN = mesenteric lymph node.

The online version of this article includes the following source data and figure supplement(s) for figure 4:

**Source data 1.** CFU, LCN2 ELISA, and ampicillin quantification data for *Figure 4*.

**Figure supplement 1.** *Escherichia coli* population sizes and total *Salmonella enterica* serovar Typhimurium (*S*.Tm) population sizes in tissue reservoirs of mice in *Figure 4*.

**Figure supplement 1—source data 1.** CFU data for *Figure 4—figure supplement 1*.

pathogen that utilizes inflammation to bloom in the gut (*Stecher et al., 2007*), using an ELISA for fecal lipocalin-2, inflammation was detected in mice with pESBL *E. coli*, and reached particularly high levels in mice where re-seeding of *S*.Tm occurred early during the experiment (i.e. >10³ lipocalin-2 ng/g feces; *Figure 4C*). Interestingly, low-grade inflammation was detected in the absence of gut luminal *S*.Tm (*Figure 4C*) and could suggest that systemic *S*.Tm may contribute to low-grade gut inflammation, propagated by gut luminal *E. coli*. To confirm that re-seeding of *S*.Tm was dependent on lowering the concentration of fecal ampicillin by pESBL *E. coli*, we measured the concentration of fecal ampicillin using mass spectrometry (*Figure 4D*). In the presence of *E. coli* pESBL, fecal ampicillin concentrations plummeted below the minimum inhibitory concentration (MIC; detected in vitro *Bakkeren et al., 2019*) of *S*.Tm (*Figure 4D*), likely explaining the re-seeding dynamics. Thus, donor-mediated antibiotic degradation may protect the re-seeding recipients and thereby promote pESBL transfer. Importantly, as in the previous experiments, *S*.Tm transconjugants formed new plasmid reservoirs in host tissues, primarily in the mLN (*Figure 4E*). This indicates that ESBL plasmids promote their spread in host-associated Enterobacteriaceae in two different ways: by lowering local antibiotic concentrations that ensure survival of potential recipients and by subverting tissue-lodged pathogens (persisters and growing populations alike) to form long-term reservoirs within host tissues.

## Discussion

Our results suggest that plasmids can leverage facultative intracellular enteropathogenic bacteria to promote both the formation of plasmid reservoirs in host tissues and the spread of plasmids into and among gut luminal microbes passing through a given host. In our experiments using *S*.Tm infection in mice, repeated cycles of pathogen invasion and tissue reservoir formation followed by re-seeding the gut lumen allow plasmids to form long-term reservoirs in the host's tissues (*Figure 5*, **steps 1–5**). These reservoirs act as a record of gut luminal horizontal gene transfer, here shown by conjugation (*Figures 1–4*). By assessing the genetic element's tissue reservoir formation, our current study significantly extends the ecological implications of our previous work where we had focused on persisters, showing that these antibiotic persisters can promote the release of donors into the gut lumen (*Bakkeren et al., 2019*). Here, we show that tissue reservoirs can influence plasmid dynamics also without selection for persister survival in antibiotic treated tissues (*Figure 5*, **step 1**). Most likely, tissue reservoir-promoted plasmid transfer dynamics will happen during persistent infection involving phenotypically susceptible pathogen cells, provided multiple strains co-occur within a host. Since a natural ecological succession of Enterobacteriaceae has been documented in the gut (*León-Sampedro et al., 2021*; *Martinson et al., 2019*), an overarching re-seeding-conjugation-re-invasion cycle could result in the distribution of different alleles in the various strains colonizing a host. The host tissue reservoirs would dramatically expand the time scale of such transfer thereby enabling horizontal gene transfer even when the initial donor and the later recipient have never 'co-existed' within the host. Such co-existence-independent gene exchange could pertain to resistance genes or virulence

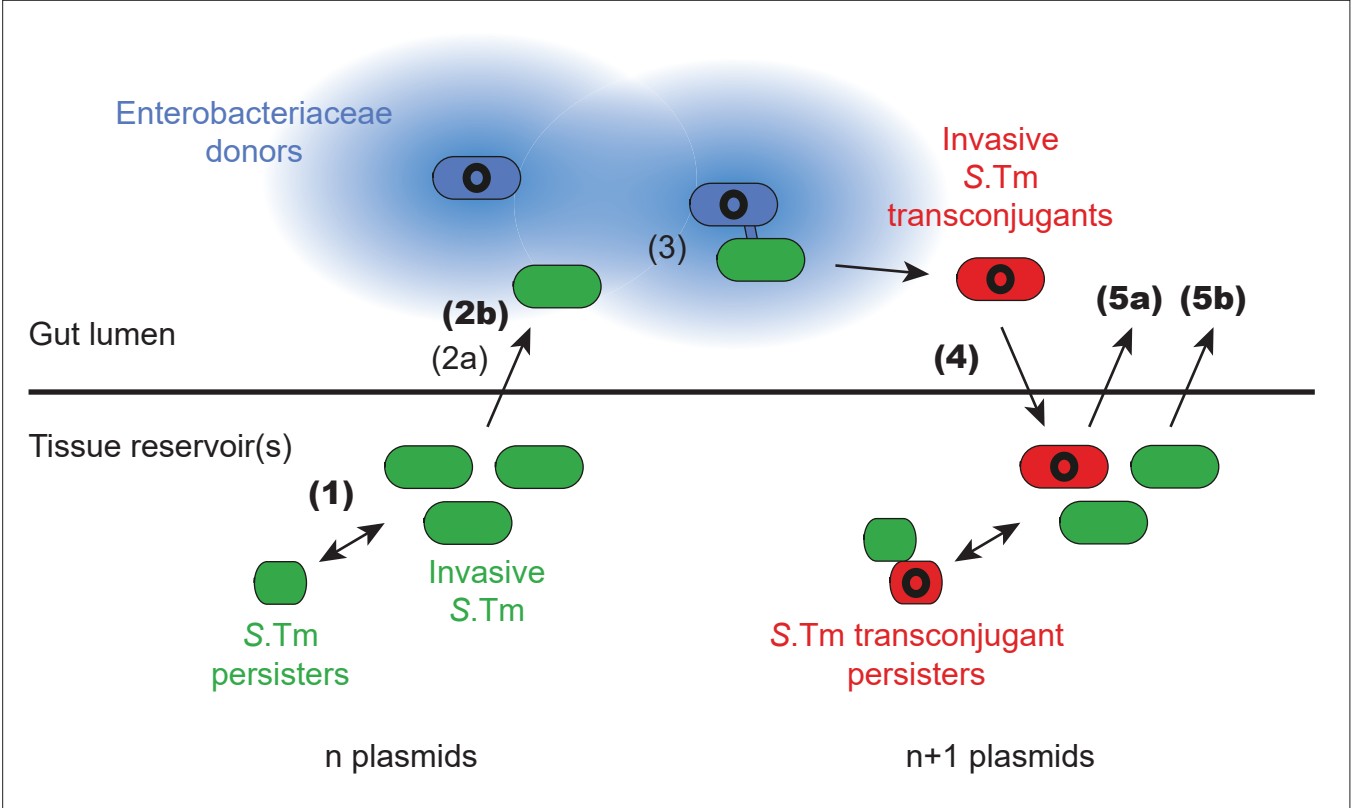

**Figure 5.** Working model for how the infection cycle of *Salmonella enterica* serovar Typhimurium (*S*.Tm) may promote the spread and accumulation of plasmids within a host. Invasive *S*.Tm (green for recipients, red for transconjugants) can establish reservoirs within host tissues, storing their current plasmid complement in these reservoirs (n plasmids). A subpopulation of these cells can survive antibiotic treatment as persisters (smaller green or red cells), which can regrow after the withdrawal of antibiotics. In extension to previous work (*Bakkeren et al., 2019*), here we show that reservoirs of *S*.Tm that chronically infect the host can re-seed from host tissues, even in the absence of selection for persisters (**step 1**). Exiting the host tissues into the gut lumen is a rate-limiting process (**step 2a**; *Bakkeren et al., 2019*). This process can be facilitated by Enterobacteriaceae donors (blue; plasmid shown in black) that produce extracellularly acting plasmid-encoded antibiotic resistance genes (such as beta-lactamases; blue colour radiating from the cell indicates local degradation of beta-lactam antibiotics by beta-lactamases), which can allow the local survival of re-seeding recipients under antibiotic treatment (**step 2b**). Once both donors and recipients co-occur in the gut lumen at sufficient density, plasmid transfer proceeds (**step 3**; *Bakkeren et al., 2019*; *Benz et al., 2021*; *Diard et al., 2017*; *Moor et al., 2017*; *Stecher et al., 2012*; *Wotzka et al., 2019*). When colonization resistance is relaxed, invasive transconjugants (red) can reach sufficient densities to invade back into host tissues and form new reservoirs that now contain n + 1 plasmids (**step 4**). In the tissue, transconjugant subpopulations can also survive antibiotic therapy as persisters (smaller red cells). This process is repeatable in certain conditions. Either transconjugants can re-seed to donate the plasmid to further gut luminal recipients (step 5a), or recipients can re-seed to receive an additional plasmids (**step 5b**). Novel steps of this process, demonstrated here using the streptomycin treatment mouse model, are indicated with bold numbers, whereas steps that have been previously shown are indicated with non-bold numbers (and key references are listed in the caption).

determinants alike. In line with this hypothesis, different strains of commensal and pathogenic Enterobacteriaceae that show vastly different plasmid, prophage, and mutational profiles are often observed within a given host population or even within the same host (*Desai et al., 2013*; *Kroger et al., 2012*; *Martinson et al., 2019*; *San Román et al., 2018*; *Tenaillon et al., 2010*). Conversely, the same plasmid has been detected in different enterobacterial strains over time within the same patient (*León-Sampedro et al., 2021*). The role of the pathogen's re-seeding-acquisition-re-invasion cycles in resistance plasmid transfer may be particularly important in cases where different *S*.Tm are present within the same host (e.g. the gut lumen vs. the host tissues), as observed in pig farms (*San Román et al., 2018*).

Which factors limit plasmid storage and transfer from host-tissue reservoirs? Through population dynamic analysis, we showed that re-seeding of recipient strains from tissue reservoirs is the rate-limiting step for the formation of new transconjugant reservoirs (*Figure 2*; *Figure 5*, step 2a). This is in line with our previous work (*Bakkeren et al., 2019*) where we suggested that donor-re-seeding from persister reservoirs in host tissues is a rate limiting process, and reinforces the importance of

understanding the mechanisms of re-seeding, host tissue invasion, and the formation of new tissue reservoirs. Previous studies have suggested the gall bladder as a route of re-seeding for *Salmonella* via bile secretions (*Everest et al., 2001*; *Gonzalez-Escobedo et al., 2011*; *Lam and Monack, 2014*). In our experimental model, we found that sequence tags in the gall bladder did not necessarily correlate with those found at highest density in the feces, suggesting, along with others (*Lam and Monack, 2014*), that additional routes of re-seeding may exist. Phagocytic cells such as macrophages and dendritic cells serve as reservoirs for *S*.Tm (*Claudi et al., 2014*; *Helaine et al., 2014*; *Kaiser et al., 2014*; *Pham et al., 2020*; *Rosenberg et al., 2021*; *Stapels et al., 2018*) and can be reactivated by IFNγ depletion (*Monack et al., 2004*). The high density of *S*.Tm in the tissues that line S.I. GALT, correlating with *S*.Tm found in the gut lumen of the small intestine, could suggest that trafficking of *S*.Tm-bearing immune cells to these sites (*Habtezion et al., 2016*) contributes to gut luminal re-seeding. The independence of *S*.Tm on its TTSS-1 for re-seeding appears to support host-mediated trafficking to the gut lumen rather than an active mechanism evolved by *S*.Tm, but further work is needed for confirmation.

Interventions such as vaccination have shown success in reducing the formation of tissue reservoirs and consequently plasmid transfer dynamics associated with re-seeding (*Bakkeren et al., 2019*; *Moor et al., 2017*; *Moor et al., 2016*). Here, we showed that once re-seeding occurred, the transfer of plasmids followed by re-invasion is likely, given a sufficiently high carrying capacity (as a proxy for decreased colonization resistance) in the gut lumen. This is because plasmid transfer is dependent on both the density of donors and recipients. Invasion may also be density-dependent (*Ackermann et al., 2008*); approximately 300 invasion events occur into the mLN per day initially in streptomycin-pretreated mouse models (*Kaiser et al., 2013*) and this likely decreases proportionally with colonization. In a healthy microbiota community, colonization resistance prohibits enterobacteriaceal blooms in >90% of the hosts (*Stecher et al., 2010*; *Stecher and Hardt, 2011*) and dramatically reduces the efficiency of plasmid transfer (*Stecher et al., 2012*; *Wotzka et al., 2019*). However, gut perturbations such as diet shifts, inflammation, antibiotic treatment, or a reduction in microbiota complexity (e.g. dysbiosis) can all lead to increases in loads of Enterobacteriaceae and consequently plasmid transfer (*Figure 5*, step 3) (*Bakkeren et al., 2019*; *Barthel et al., 2003*; *Benz et al., 2021*; *Kreuzer and Hardt, 2020*; *Stecher et al., 2012*; *Stecher et al., 2007*; *Wotzka et al., 2019*). Therefore, a healthy microbiota is integral in minimizing pathogen evolution in two complementary ways, by reducing horizontal gene transfer and by reducing the likelihood of re-invasion and subsequent formation of tissue reservoirs. Further work will be needed to validate how such perturbations accelerate the dynamics of plasmid spread and reservoir formation.

Lastly, we showed that plasmids encoding secreted antibiotic degrading enzymes can promote their own spread by an additional mechanism in vivo: protecting potential new recipients sensitive to the antibiotic (*Figure 5*, **step 2b**). Re-seeding of *S*.Tm, initially susceptible to the beta-lactam, can occur even during beta-lactam therapy, dependent on the presence of *E. coli*-producing beta-lactamase (*Figure 4*). The prevalence of ESBL-producing *E. coli* is rising in recent years, and ESBL genes are frequently plasmid-encoded (*Temkin et al., 2014*; *World Health Organization, 2017*; *Yang et al., 2009*). The extracellularly acting beta-lactamases are therefore public goods. In vitro this was shown to allow the survival of susceptible *S*.Tm in the presence of otherwise lethal concentrations of ampicillin (*Perlin et al., 2009*). In our experiments, we showed that the same process occurs in the gut lumen, with re-seeding *S*.Tm followed by conjugation, clonal expansion, and tissue re-invasion as a consequence (*Figure 4*). As one of the prodigal mechanisms for blooms (of resistant bacteria), antibiotic treatment could serve to amplify re-seeding followed by plasmid exchange and tissue reservoir reformation, given sufficient protection from the antibiotic in trans. This obviously depends on the presence of tissue-invasive enteropathogens in the hosts, which is quite common in today's farming industry (*San Román et al., 2018*; *Van Boeckel et al., 2019*), as well as the mechanism of action of the antibiotic used. Many antibiotic resistance genes encode antibiotic-inactivating enzymes, even if they are not secreted (*D'Costa et al., 2006*), as is the case of beta-lactamases. However, it remains to be seen to what extent non-secreted inactivating enzymes can deplete local concentrations of the antibiotic in vivo. Conversely, pathogen intrinsic mechanisms transiently promoting survival may further prolong recipient survival in an antibiotic loaded gut lumen and thereby further promote plasmid spread. The transient overexpression of efflux pumps was shown to confer survival long enough to obtain a plasmid in the presence of tetracycline above the MIC in vitro (*Nolivos et al.,*

*2019*). Moreover, bacterial strains that produce outer membrane vesicles have been shown to lead to an altered susceptibility to antimicrobial peptides (e.g. polymyxin B or colistin, shown in several bacteria including *E. coli* and *Salmonella* Typhi) (*Manning and Kuehn, 2011*; *Marchant et al., 2021*). One study found that polymyxin resistance could even be functionally transferred to sensitive bacteria mediated by outer membrane vesicles (*Marchant et al., 2021*). It is therefore plausible that mechanisms aside from secreted and diffusible enzymes (such as for beta-lactamases) could contribute to re-seeding and plasmid acquisition. However, further work will be necessary to determine different possible mechanisms for cross-protection of susceptible strains by resistant cells in the gut lumen.

Overall, here we show that plasmids can subvert tissue-associated pathogen reservoirs to promote long-term carriage within the host and subsequent spread. There seem to be two types of tissue reservoirs with partially complementary features (*Figure 5*). The tissue-associated reservoirs formed by enteropathogens in chronic (persistent) infections are relatively large and facilitate re-seeding of the gut lumen and subsequent plasmid transfer at relatively high rates. In contrast, persistent pathogen cells are less frequent and need to re-start growth before engaging in any luminal plasmid transfers, likely making them less efficient at luminal re-seeding. However, they ensure survival of recipients and plasmid reservoir maintenance even in cases of therapy with antibiotics to which these persisters are genetically susceptible. Although in our experimental model plasmids can benefit strongly from tissue reservoirs, their actual transfer is dependent on the strain densities in the gut lumen. Our study provides a proof-of-concept that tissue reservoirs can act as a record of evolutionary events such as plasmid transfer in the gut lumen. It is conceivable that mutational events or transfer of other mobile genetic elements, for example, bacteriophages carrying accessory genes, could also be stored in tissue reservoirs. Regardless, it remains to be seen how often such cases occur in nature, since our work identifies these processes to be dependent on the carrying capacity in the gut and conjugation rates of plasmid-strain pairs, and to what extent this contributes to the dissemination of accessory genes on mobile genetic elements, such as antibiotic resistance genes.

# Materials and methods

## Key resources table

| Reagent type (species) or resource | Designation | Source or reference | Identifiers | Additional information |
|---|---|---|---|---|
| Strain, strain background (*Mus musculus*) | 129S6/SvEvTac | Taconic Biosciences | RRID:IMSR_TAC:129sve | Wild-type mouse; specified opportunistic pathogen-free mice bred under hygienic conditions in the ETH Phenomics Center |
| Strain, strain background (*Salmonella enterica* serovar Typhimurium) | SL1344 | *Hoiseth and Stocker, 1981* | | Derivatives of this strain used in this study are listed in *Table 1* |
| Strain, strain background (*Salmonella enterica* serovar Typhimurium) | ATCC 14028S | *Jarvik et al., 2010* | | Derivatives of this strain used in this study are listed in *Table 1* |
| Strain, strain background (*Escherichia coli*) | *E. coli* 536 | *Berger et al., 1982*; *Brzuszkiewicz et al., 2006* | | Derivatives of this strain used in this study are listed in *Table 1* |
| Strain, strain background (*Escherichia coli*) | *E. coli* ESBL15 | *Bakkeren et al., 2019*; *Tschudin-Sutter et al., 2016* | | Derivatives of this strain used in this study are listed in *Table 1* |
| Recombinant DNA reagent | *TAG1-7 Cm^R or Kan^R* | *Grant et al., 2008* | | Barcodes used for population dynamics analyses |

*Continued on next page*

*Continued*

| Reagent type (species) or resource | Designation | Source or reference | Identifiers | Additional information |
|---|---|---|---|---|
| Recombinant DNA reagent | P2$^{cat}$ | *Stecher et al., 2012* | | Conjugative plasmid labelled with chloramphenicol resistance |
| Sequence-based reagents | RT-qPCR primers | *Grant et al., 2008* | | See *Table 3* |
| Sequence-based reagents | PCR primers for strain construction and validation | This study; *Bakkeren et al., 2021a* | | See *Table 3* |
| Software, algorithm | GraphPad Prism | GraphPad Prism (https://graphpad.com) | RRID:SCR_015807 | Version 8 for Windows |
| Software, algorithm | R Project for Statistical Computing | https://www.r-project.org/ | RRID:SCR_001905 | |

## Strains, plasmids, and primers used in this study

Bacterial strains used in this study are derivatives of *S*.Tm SL1344 (*Hoiseth and Stocker, 1981*), *S*.Tm ATCC 14028S (*Jarvik et al., 2010*), *E. coli* 536 (*Berger et al., 1982*; *Brzuszkiewicz et al., 2006*), or *E. coli* ESBL15 (*Bakkeren et al., 2019*; *Tschudin-Sutter et al., 2016*), and are listed in *Table 1*. For cultivation of bacterial strains, lysogeny broth (LB) medium was used containing the appropriate antibiotics (50 µg/ml streptomycin [AppliChem]; 50 µg/ml kanamycin [AppliChem]; 15 µg/ml chloramphenicol [AppliChem]; 100 µg/ml ampicillin [AppliChem]) at 37°C (or 30°C if containing pCP20 or pCP20-IncI1). Genetic constructs (e.g. gene deletions, neutral isogenic sequence tags, or the P3 plasmid) were introduced into the desired background strain using P22 HT105/1 *int-201* phage transduction (*Sternberg and Maurer, 1991*). Bacterial plasmids used to confer resistance or for construction of strains are listed in *Table 2* and were transformed into cells using electroporation.

 *E. coli* ESBL 15 was cured of its plasmid, pESBL, using plasmid incompatibility. To create an incompatible plasmid, pCP20-IncI1 was cloned using the replicon of pESBL. A PCR amplicon of the replication initiation protein of pESBL with 831 and 420 bp upstream and downstream flanking regions, respectively (to ensure the entire replicon was cloned), was cloned into pCP20 digested with *Pst*I and *Eco*RV (removing the ampicillin resistance cassette) using Gibson assembly (NEB; protocol as described by the manufacturer). The resulting plasmid, pCP20-IncI1, was electroporated into *E. coli* ESBL15 and grown at 30°C under chloramphenicol selection. Loss of resistance of ampicillin (and consequently loss of pESBL) was confirmed by streaking on LB with ampicillin. The resulting clones were restreaked on LB without chloramphenicol grown at 37°C to cure pCP20-IncI1.

 Conjugative plasmids were transferred into the desired strain (e.g. P2$^{cat}$ into *E. coli* ΔpESBL and *E. coli* 536) using in vitro conjugation assays. In brief, $10^5$ CFU from an overnight culture of the donor strain (SL1344 P2$^{cat}$) was mixed with the desired recipient, allowing conjugation overnight at 37°C on a rotating wheel. Cells were plated on selective MacConkey agar to identify transconjugants by resistance phenotype and/or colour (i.e. *S*.Tm is *lac* negative and thus forms yellow colonies on MacConkey agar while *E. coli* is *lac* positive and forms red colonies).

 All strains and plasmids were genotyped prior to use using the primers listed in *Table 3*.

## Infection experiments

All animal experiments were performed in 8- to 12-week-old specified opportunistic pathogen-free 129/SvEv mice. These mice contain a functional *Nramp1* allele (also known as *Slc11a1*) and are resistant to *S*.Tm and therefore allow for long-term infections (*Cunrath and Bumann, 2019*; *Stecher et al., 2006*). All infection experiments were approved by the responsible authorities (Tierversuchskommission,

**Table 1.** Strains used in this study.

| Strain name | Strain number | Relevant genotype | Resistance[*] | Reference |
|---|---|---|---|---|
| SL1344 | SB300 | Wild-type | Sm | *Hoiseth and Stocker, 1981* |
| ATCC 14028S | 14028S | Wild-type | None | *Jarvik et al., 2010* |
| SL1344 P2$^{cat}$ | M995 | *cat* on P2 | Sm, Cm | *Stecher et al., 2012* |
| SL1344 ΔP2 | M1404 | P2 cured | Sm | *Stecher et al., 2012* |
| *E. coli* 536 | Z2366 | Wild-type | Sm | *Berger et al., 1982*; *Brzuszkiewicz et al., 2006* |
| *E. coli* 536 P2$^{cat}$ | Z2124 | *cat* on P2 | Sm, Cm | This study |
| SL1344 ΔP2 pM975 | Z2287 | P2 cured; *bla* | Sm, Amp | This study |
| *E. coli* ESBL15 | Z2115 | *CTX-M1 on pESBL* | Amp | *Bakkeren et al., 2019*; *Tschudin-Sutter et al., 2016* |
| 14028S Sm$^R$ | T186 | *strAB* on P3 | Sm | This study |
| 14028S Sm$^R$ TAG1 | Z2279 | *WITS1-aphT* | Sm, Kan | This study |
| 14028S Sm$^R$ TAG2 | Z2281 | *WITS2-aphT* | Sm, Kan | This study |
| 14028S Sm$^R$ TAG3 | Z2283 | *WITS11-aphT* | Sm, Kan | This study |
| 14028S Sm$^R$ TAG4 | T270 | *WITS13-aphT* | Sm, Kan | This study |
| 14028S Sm$^R$ TAG5 | T272 | *WITS17-aphT* | Sm, Kan | This study |
| 14028S Sm$^R$ TAG6 | T274 | *WITS19-aphT* | Sm, Kan | This study |
| 14028S Sm$^R$ TAG7 | T276 | *WITS21-aphT* | Sm, Kan | This study |
| 14028S$^{SPI-2}$ Sm$^R$ | T284 | *ssaV::aphT* | Sm, Kan | This study |
| SL1344$^{Avir}$ P2$^{TAG4}$ | Z2292 | *WITS13-cat* on P2; Δ*invG* Δ*ssaV* | Sm, Cm | This study |
| SL1344$^{Avir}$ P2$^{TAG5}$ | Z2293 | *WITS17-cat* on P2; Δ*invG* Δ*ssaV* | Sm, Cm | This study |
| SL1344$^{Avir}$ P2$^{TAG6}$ | Z2294 | *WITS19-cat* on P2; Δ*invG* Δ*ssaV* | Sm, Cm | This study |
| SL1344$^{Avir}$ P2$^{TAG7}$ | Z2295 | *WITS21-cat* on P2; Δ*invG* Δ*ssaV* | Sm, Cm | This study |
| *E. coli* ΔpESBL | Z2156 | pESBL cured | None | This study |
| *E. coli* ΔpESBL P2$^{cat}$ | T305 | pESBL cured; *cat* on P2 | Cm | This study |
| 14028S$^{SPI-1}$ Sm$^R$ | T2429 | *invG::aphT* | Sm, Kan | This study |

[*]Relevant resistances only: Sm = ≥50 µg/ml streptomycin; Cm = ≥15 µg/ml chloramphenicol; Kan = ≥50 µg/ml kanamycin; Amp = ≥100 µg/ml ampicillin.

Kantonales Veterinäramt Zürich, licenses 193/2016 and 158/2019). Sample size was not predetermined and mice were randomly assigned to treatment groups.

For all experiments, overnight cultures of recipient *S*.Tm (14028S Sm$^R$ derivatives) containing the appropriate antibiotics were washed with sterile PBS 2× before being diluted to $10^4$ CFU/ml in PBS. Mixtures of the appropriate strains (in the case of tagged strains) or single strains were intraperitoneally injected into mice as a 100 µl volume (~$10^3$ CFU per mouse). All mice used for plasmid transfer experiments were housed in individual cages to ensure experimental independence. For experiments that lasted longer than 10 days, mice were caged in pairs initially, but split to individual cages for the final 10 days (due to ethical reasons, mice were not individually caged for longer than 10 days).

On day 2 post infection, mice were given an oral dose of streptomycin (20 mg), as previously described for the streptomycin-pretreated mouse model (*Barthel et al., 2003*). Overnight cultures

**Table 2.** Plasmids used in this study.

| Plasmid name | Relevant genotype | Resistance | Reference |
|---|---|---|---|
| pM975 | *bla*; used to confer ampicillin resistance | Amp | *Hapfelmeier et al., 2005* |
| pCP20 | FLP recombinase | Amp, Cm | *Datsenko and Wanner, 2000* |
| pCP20-Incl1 (pZ2496) | FLP-*bla*::*Incl*1 replicon | Cm | This study |
| P3 (pRSF1010) | *strAB* | Sm | *Kroger et al., 2012* |
| P2 (pCol1b9) | Wild-type | None | *Stecher et al., 2012* |
| P2$^{cat}$ | *cat* | Cm | *Stecher et al., 2012* |
| P2$^{TAG}$ | *WITS13, 17, 19,or 21-cat* on P2 | Cm | *Bakkeren et al., 2019* |
| pESBL (pESBL15) | *CTX-M-1* | Amp | *Bakkeren et al., 2019*; *Tschudin-Sutter et al., 2016* |

of donor strains (Sm$^R$; SL1344 or *E. coli* 536 derivatives) containing the appropriate antibiotics were subcultured for 4 hr 1:20 in 2 ml LB without antibiotics, and then washed in PBS. Four hours after streptomycin treatment, donors were given orally (~5 × 10$^7$ CFU per mouse). After 10 days, mice were either euthanized or given doses of oral ciprofloxacin (3 mg per mouse; ciprofloxacin hydrochloride monohydrate, Sigma-Aldrich, dissolved in 100 µl sterile dH$_2$O) for 3 consecutive days, while ampicillin (2 g/l) was provided in the drinking water on days 11–15 to prevent re-seeding before the third strain was added. After each ciprofloxacin treatment, cages were changed to prevent reinfection due to coprophagy. On day 15 of the experiment, mice were either euthanized or given a third strain (either a second recipient or a second donor; Amp$^R$). These mice were euthanized on day 22. For *Figure 3—figure supplement 3*, mice were not i.p. infected with 14028S, so the experiment started with an oral streptomycin treatment (20 mg) 4 hr prior to infection with *E. coli* 536 P2$^{cat}$. The rest of the experiment was treated the same as *Figure 3*, but after 4 days of ampicillin in the drinking water (as above), the mice were switched to streptomycin drinking water (1 g/l) for 3 days before euthanasia.

For mice used to enrich for persisters in tissue reservoirs (*Figure 1—figure supplement 5*), ceftriaxone (1.5 mg ceftriaxone disodium salt hemi(heptahydrate) dissolved in 100 µl PBS; Sigma-Aldrich) was injected intraperitoneally for 3 consecutive days. Donors were added after a streptomycin treatment (as above) on day 6 and again without streptomycin on day 8 to ensure robust colonization. These mice were euthanized on day 15.

For mice used to analyse plasmid transfer within tissues (*Figure 3—figure supplement 4*), overnight cultures of SL1344 P2$^{cat}$ and 14028S Sm$^R$ Kan$^R$ were washed with sterile PBS 3× before being diluted to 10$^4$ CFU/ml in PBS; 50 µl of each strain was injected into the same side of the mouse by an i.p. injection, 5 min sequentially (total CFU ~ 10$^3$ CFU per mouse). Mice were kept for 3 days before euthanasia and analysis of organs.

For mice used to investigate re-seeding in the presence of ampicillin counterselection dependent on ESBL-producing *E. coli* (*Figure 4*), on day 1 post intraperitoneal infection of recipients, the mice were given 0.5 g/l ampicillin in the drinking water and given ~5 × 10$^7$ CFU pESBL or P2$^{cat}$ *E. coli* donors orally (in PBS, following a 4 hr subculture as described above). An ELISA for mouse lipocalin-2 was performed on feces (protocol according to the manufacturer; R&D Systems kit) to determine the inflammatory state of the gut.

In all mouse experiments, feces were collected daily into pre-weighed Eppendorf tubes and homogenized in PBS using a steel ball at 25 Hz for 1 min. Populations of donors, recipients, and transconjugants were diluted and enumerated using selective MacConkey agar. Replica plating was used to accurately determine the ratio of recipients or donors to transconjugants if transconjugants were on the same order of magnitude as the donors or recipients. When mice were euthanized (at day 2, 10, 15, or 22; specified in the figure legends), the spleen, liver, gall bladder, and mLN were collected and homogenized in PBS. Populations of bacteria were enumerated as for the feces. For mice euthanized at day 2 post infection (*Figure 1—figure supplement 6*), additional organs were collected. For the duodenum, jejunum, ileum, cecum, and colon, first the content was collected, and then 1 cm of each tissue was opened longitudinally and washed briefly in PBS before homogenization. For the

**Table 3.** Primers used in this study.

| Primer name | Sequence (5' to 3') | Purpose | Reference |
|---|---|---|---|
| ESBL15_repl_gibs_for | GCC AGT TAA TAG TTT GCG CAA CGT TGT TGC CAT TGC TGC ACT GAG CTA CCA TAG ATG AC | PCR amplification of IncI1 replicon from pEBSL for Gibson assembly | This study |
| ESBL15_repl_gibs_rev | TAC AAT TAT TCC TTA CTA TGG ACA AAA ACA TCA ATC TGA T GTG GTT TCA GAA CGG TGA G | PCR amplification of IncI1 replicon from pEBSL for Gibson assembly | This study |
| ESBL15_ori_ver_up | CCA GTT AAT AGT TTG CGC AAC | Validation of pCP20-IncI1 cloning | This study |
| ESBL15_ori_intl_ rev | CTT TCA GCG CTT TAT AGC G | Validation of pCP20-IncI1 cloning | This study |
| ESBL15_ori_intl_for | CTG TTC CGA TGA CCA TCT G | Validation of pCP20-IncI1 cloning | This study |
| ESBL15_ori_ver_dw | CTC CAG TTT AAA TAC AAG ACG | Validation of pCP20-IncI1 cloning | This study |
| ssaV-137F | GCAGCGTTCCA GGGTATTCC | Verification of ΔssaV in the chromosome | **Bakkeren et al., 2021a** |
| ssaV +155 R | CAGCAAGTTCT TCTCCAGGC | Verification of ΔssaV in the chromosome | **Bakkeren et al., 2021a** |
| invG-134F | GAAGGCCACGA GAACATCAC | Verification of ΔinvG in the chromosome | **Bakkeren et al., 2021a** |
| invG +112 R | GCGGCCTGTT GTATTTCCGC | Verification of ΔinvG in the chromosome | **Bakkeren et al., 2021a** |
| P3_SmR_for | CTA GTA TGA CGT CTG TCG C | Verification of P3 | This study |
| P3_SmR_rev | CAC GTT TCG CAA CCT GTT C | Verification of P3 | This study |
| WITS1 | ACG ACA CCA CTC CAC ACC TA | qPCR for TAG1 | **Grant et al., 2008** |

*Table 3 continued on next page*

*Table 3 continued*

| Primer name | Sequence (5' to 3') | Purpose | Reference |
|---|---|---|---|
| WITS2 | ACC CGC AAT ACC AAC AAC TC | qPCR for TAG2 | *Grant et al., 2008* |
| WITS11 | ATC CCA CAC ACT CGA TCT CA | qPCR for TAG11 | *Grant et al., 2008* |
| WITS13 | GCT AAA GAC ACC CCT CAC TCA | qPCR for TAG13 | *Grant et al., 2008* |
| WITS17 | TCA CCA GCC CAC CCC CTC A | qPCR for TAG17 | *Grant et al., 2008* |
| WITS19 | GCA CTA TCC AGC CCC ATA AC | qPCR for TAG19 | *Grant et al., 2008* |
| WITS21 | ACA ACC ACC GAT CAC TCT CC | qPCR for TAG21 | *Grant et al., 2008* |
| ydgA | GGC TGT CCG CAA TGG GTC | qPCR for all tags | *Grant et al., 2008* |
| pagJ_fwd | ATC TTC GGG GAA GGG CAC GTC CG | 14028S chromosome-specific PCR | This study |
| pagJ_rev | GCT GTA ACC GTA AGG ATA GTG TGC CAC AAT T | 14028S chromosome-specific PCR | This study |
| cat_fwd | GCATTTCAGTCA GTTGCTCAATGT ACCTATAACC | P2$^{cat}$-specific PCR | This study |
| cat_rev | CGACATGGAAGC CATCACAAACGG | P2$^{cat}$-specific PCR | This study |

purposes of this study, the small intestinal and large intestinal gut-associated lymphoid tissue (S.I. and L.I. GALT) was collected by removing the lining of the tissues (including the fat) for each of the small intestine and large intestine. For analysis of bacterial loads in the blood, 100 µl of blood was aspirated from the heart immediately after euthanasia and collected in PBS with 2% BSA and 1 mM EDTA. Blood samples were also homogenized. Dissection tools were disinfected in 70% EtOH in-between each organ to minimize the chance of cross-contamination between organs.

## Analysis of population dynamics using neutral sequence tags

Mice were infected with an equal ratio of S.Tm 14028S Sm$^R$ recipient strains (total ~$10^3$ CFU) bearing sequence tags at a neutral locus in the chromosome (*Grant et al., 2008*). For analysis of re-seeding and plasmid transfer dynamics in *Figure 2*, three recipient tagged strains were used (TAG1-3 Kan$^R$; 1:1:1 ratio). On day 2 post infection, an equal ratio of donor tagged strains (S.Tm SL1344$^{Avir}$ P2$^{TAG}$ strains; TAG4-7 Cm$^R$; 1:1:1:1) were given at a total inoculum size of ~$5 \times 10^7$ CFU per mouse orally. The inocula were enriched in either LB+ chloramphenicol (for donors) or LB+ kanamycin (for recipients). At the end of the experiment (i.e. on day 10 post infection), mice were euthanized and recipient+ transconjugant populations were enriched from 100 µl of the feces and organ homogenates (in parallel to selective plating) in LB supplemented with kanamycin.

For analysis of re-seeding dynamics in the presence or absence of donors (*Figure 1—figure supplement 4*; *Figure 2—figure supplement 2*), an equal ratio of S.Tm 14028S Sm$^R$ recipient strains (total ~$10^3$ CFU) bearing seven sequence unique tags at a neutral locus in the chromosome were used.

Enrichments were concentrated and genomic DNA was extracted using a QIAamp DNA Mini Kit (Qiagen). qPCR analysis was performed according to temperature conditions as previously described (*Grant et al., 2008*) using qPCR primers specific to the TAG and a universal second primer, *ydgA* (*Table 3*). The relative proportion was calculated by dividing the DNA copy number (calculated from the $C_T$ value) of each tag detected, by the sum of all seven tags in the sample. A dilution standard of purified chromosomal DNA allowed for a correlation between DNA copy number and $C_T$ value. For each qPCR run, the detection limit was determined by the $C_T$ value of the most-diluted DNA standard. The least precise detection limit defines the conservative detection limit plotted on graphs and used for the mathematical model ($8.9 \times 10^{-5}$). Once the relative proportion was determined, recipient tags were separated from plasmid tags (however, the proportion remains relative to all tags). This data is presented in *Figure 2—figure supplement 1*. Next, the tags were sorted according to the most abundant in the feces in each given mouse (separately for recipient and plasmid tags). This ranking was conserved for analysis in other organs to allow a correlation between the most abundant tag in the organs relative to the feces. For example, if TAG2 appeared the most abundant in the feces but least abundant in the mLN, it would ranked and coloured according to the most abundant tag, and it would still appear the same colour in the mLN, despite it not being the most abundant tag in the mLN. This data is presented in *Figure 2A–B*. The samples in *Figure 2—figure supplement 2* were analysed the same, but were not sorted according to the most abundant tag in the feces, and is instead presented as raw tag proportions relative to all seven recipient chromosomal tags.

## Measurement of fecal ampicillin

Detection and absolute quantification of ampicillin was carried out using high-performance liquid chromatography heated electrospray ionization high-resolution mass spectrometry (HPLC HESI HRMS). Fecal samples were homogenized in PBS and immediately frozen after collection. Prior to analysis, samples were thawed and centrifuged at high speed. The supernatants were analysed on a Dionex UltiMate 3000 HPLC coupled to a Q Exactive Hybrid Quadrupole-Orbitrap mass spectrometer. Chromatographic separation was obtained on a Phenomenex Kinetex 2.6 µm XB-C18 150 × 4.6 mm column at 30°C. Water (A) and acetonitrile (B), each containing 0.1% formic acid, were employed as mobile phases. A gradient of total 12.5 min was applied at a flowrate of 800 µl/min, starting with 3% B for 2 min, 3–50% B in 3 min, 50–98% B in 5 min, 98% B for 1 min, 98–3% B in 1 min and 3% B for 0.5 min. MS settings: spray voltage (+) 3.5 kV, capillary temperature 320°C; sheath gas (57.50), auxiliary gas (16.25), sweep gas (3.25); probe heater 462.50°C; S-Lens RF (50), resolution (70.000); AGC target (3e6), microscans (1), maximum IT 200 ms, scan range 250–750 *m/z*. For quantification, an ampicillin standard curve was recorded using H$_2$O diluted concentrations of 10 ng/ml, 100 ng/ml, 500 ng/ml, 1 µg/ml, 10 µg/ml, and 100 µg/ml from a 1 mg/ml ampicillin stock solution. Samples were

analysed by retention time of ampicillin and the respective ion adduct $[M + H]^+ = 350.1169$ $m/z$ with a mass tolerance of 5 ppm; 15 µl of each sample were injected and ampicillin concentrations were calculated using the Thermo Xcalibur 2.2 Quan Browser software. The concentration given in µg/ml of supernatant was normalized to the weight of feces collected and converted to µg/ml fecal content using the average density of feces (*Brown et al., 1996*).

## Statistical analysis

Statistical tests on experimental data were performed using GraphPad Prism 8 for Windows. The specific test used is each figure is described in the figure caption (a test is only indicated if statistics were used). The mathematical model was fit to the experimental measurements using an approximate Bayesian computation (ABC) approach (*Marjoram et al., 2003*). For the fit, we considered three sets of summary statistics: the skew of the plasmid and chromosomal tag distributions, the total size of the transconjugant and recipient population on day 10, and the time at which the transconjugant and recipient populations first exceeded $10^6$ CFU/g feces. A simulation was called 'successful' if all summary statistics were within three standard deviations of the experimentally observed mean of these statistics. Migration and conjugation rates were varied on a grid, all other parameters were kept fixed (*Appendix 1—table 1*). All R-code needed to simulate the stochastic model, estimate the most likely parameters from the experimental data, and plot the results is included in the Github repository (https://github.com/JSHuisman/Recorder; *Bakkeren, 2021b* copy archived at swh:1:rev:2822d696ceddeca01a2d3eb32ffcc9bd513e561a).

## Acknowledgements

We would like to thank the staff at RCHCI and EPIC animal facilities, and members of the Hardt, Sunagawa, and Slack labs for helpful discussions. We would also like to acknowledge the reviewers for their constructive feedback and helpful suggestions to improve this manuscript.

## Additional information

### Funding

| Funder | Grant reference number | Author |
|---|---|---|
| Schweizerischer Nationalfonds zur Förderung der Wissenschaftlichen Forschung | 407240_167121 | Sebastian Bonhoeffer Wolf-Dietrich Hardt |
| Schweizerischer Nationalfonds zur Förderung der Wissenschaftlichen Forschung | 3100030B_173338 | Wolf-Dietrich Hardt |
| Schweizerischer Nationalfonds zur Förderung der Wissenschaftlichen Forschung | 310030_192567 | Wolf-Dietrich Hardt |
| Gebert Rüf Stiftung | GRS-060/18 | Wolf-Dietrich Hardt |
| Monique Dornonville de la Cour Foundation | | Wolf-Dietrich Hardt |
| Schweizerischer Nationalfonds zur Förderung der Wissenschaftlichen Forschung | PP00PP_176954 | Mederic Diard |
| Botnar Research Centre for Child Health | Multi-investigator grant | Mederic Diard |

| Funder | Grant reference number | Author |
| --- | --- | --- |
| Boehringer Ingelheim Fonds | PhD Fellowship | Erik Bakkeren |

The funders had no role in study design, data collection and interpretation, or the decision to submit the work for publication.

## Author contributions

Erik Bakkeren, Conceptualization, Data curation, Formal analysis, Investigation, Methodology, Project administration, Resources, Supervision, Validation, Visualization, Writing – original draft, Writing – review and editing; Joana Anuschka Herter, Data curation, Formal analysis, Investigation, Methodology, Resources, Validation, Writing – review and editing; Jana Sanne Huisman, Data curation, Formal analysis, Investigation, Methodology, Resources, Software, Validation, Writing – review and editing; Yves Steiger, Investigation, Resources, Writing – review and editing; Ersin Gül, Investigation, Methodology, Validation, Writing – review and editing; Joshua Patrick Mark Newson, Investigation, Writing – review and editing; Alexander Oliver Brachmann, Formal analysis, Investigation, Methodology, Validation, Writing – review and editing; Jörn Piel, Funding acquisition, Resources, Supervision, Writing – review and editing; Roland Regoes, Sebastian Bonhoeffer, Médéric Diard, Funding acquisition, Supervision, Writing – review and editing; Wolf-Dietrich Hardt, Conceptualization, Funding acquisition, Project administration, Resources, Supervision, Writing – review and editing

## Author ORCIDs

Erik Bakkeren ⓘ http://orcid.org/0000-0001-7970-7890
Jana Sanne Huisman ⓘ http://orcid.org/0000-0002-1782-8109
Ersin Gül ⓘ http://orcid.org/0000-0003-2873-8034
Joshua Patrick Mark Newson ⓘ http://orcid.org/0000-0003-2091-7943
Roland Regoes ⓘ http://orcid.org/0000-0001-8319-5293
Sebastian Bonhoeffer ⓘ http://orcid.org/0000-0001-8052-3925
Médéric Diard ⓘ http://orcid.org/0000-0003-0851-4391
Wolf-Dietrich Hardt ⓘ http://orcid.org/0000-0002-9892-6420

## Ethics

All infection experiments were approved by the responsible authorities (Tierversuchskommission, Kantonales Veterinäramt Zürich, licenses 193/2016 and 158/2019).

## Decision letter and Author response

Decision letter https://doi.org/10.7554/eLife.69744.sa1
Author response https://doi.org/10.7554/eLife.69744.sa2

# Additional files

## Supplementary files

• Transparent reporting form

## Data availability

All data generated or analysed during this study are included in the manuscript and supporting files. Source data, code, and simulation results have been provided for all figures either in the article or at the following GitHub repository: https://github.com/JSHuisman/Recorder, copy archived at https://archive.softwareheritage.org/swh:1:rev:2822d696ceddeca01a2d3eb32ffcc9bd513e561a.

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

## Appendix 1

### Description of the mathematical model

We model the re-seeding of *Salmonella* recipients from systemic sites into the gut, and the subsequent creation of transconjugants through plasmid transfer from donors. We follow the dynamics of donor (D), recipient (R), and transconjugant (T) populations in the gut. These populations are further divided into isogenic subpopulations based on their sequence tags. The donors $D_j$ carry a plasmid tag $j \in 1, ..., N_j$, where $N_j$ is the number of plasmid tags ($N_j = 4$). Recipients $R_i$ carry a chromosomal tag $i \in 1, ..., N_i$, where $N_i$ is the total number of chromosomal tags ($N_i = 3$). Transconjugants $T_{ij}$ carry both a chromosomal and a plasmid tag.

To describe the dynamics of these populations, we make several assumptions. Recipients are introduced into the gut at an overall rate μ. Here, we assume this rate is constant, although it likely increases slightly during the experiment as a function of the recipient population size at systemic sites. We assume the populations of recipients and donors in the gut are well mixed, and plasmid transfer is described by mass action kinetics. Plasmids are thus transferred between donors and recipients at a constant per-contact rate $\gamma_D$. Once transconjugants are formed, these can also transfer their plasmid at a constant per-contact rate $\gamma_T$. All plasmids are isogenic; they transfer from donors or transconjugants at the same rate and can be transferred only to naïve recipients.

Resources are limited in the gut, so we model bacterial population growth as a logistic function of the population size. We assume the donor population is limited by a different resource than the recipient and transconjugant populations, since they are not isogenic (*Lee et al., 2013*). The populations thus reach separate carrying capacities, denoted by $K_D$ and $K_{RT}$, respectively. Otherwise, the growth dynamics are assumed population-independent: the bacteria are born at a birth-rate r, are cleared at a clearance rate c (this includes both death and efflux from the gut), and similar to *Bakkeren et al., 2019*; we parametrize this with a residual birth-rate at carrying capacity $r_K = c$ (to ensure some continued population turnover even at high densities).

We simulate the population dynamics stochastically using the tau-leaping method from the R package *adaptivetau* (*Johnson, 2016*). The corresponding deterministic equations (in the limit of large numbers) are:

$$\dot{D_j} = (r + r_K)\, D_j - D_j \left( c + r \frac{\sum_j D_j}{K_D} \right) \tag{1}$$

$$\dot{R_i} = (r + r_K)\, R_i + \tfrac{1}{N_i}\mu - \sum_j \left( \gamma_D D_j + \gamma_T \sum_i T_{ij} \right) R_i - R_i \left( c + r \frac{\sum_i (R_i + \sum_j T_{ij})}{K_{RT}} \right) \tag{2}$$

$$\dot{T_{ij}} = (r + r_K)\, T_{ij} + \left( \gamma_D D_j + \gamma_T \sum_i T_{ij} \right) R_i - T_{ij} \left( c + r \frac{\sum_i (R_i + \sum_j T_{ij})}{K_{RT}} \right) \tag{3}$$

### Parametrizing the mathematical model

In the main simulations, we varied the migration rate μ and the conjugation rates $\gamma_{D/T}$ while the other parameters were kept fixed (summarized in *Appendix 1—table 1*). The population-independent parameters describing the bacterial growth dynamics (r, c, $r_K$), as well as the carrying capacities $K_{D/RT}$ and inoculum density $D_0$, were set to the values from *Bakkeren et al., 2019*. There they were parametrized based on experimental data from the same mouse model system (*Barthel et al., 2003*; *Moor et al., 2017*). The number of distinguishable chromosomal and plasmid tag populations matches those used in this paper. We fixed the conjugation rates from donors and transconjugants to the same value, and the carrying capacities of donors and recipients + transconjugants were set to the same value.

*Appendix 1—table 1 Continued on next page*

**Appendix 1—table 1.** Parameter values used in the simulations.

| Parameter | Function | Value | Units |
|---|---|---|---|
| r | Birth-rate | 44 ln(2) | Per day |
| c | Clearance-rate | 4 ln(2) | Per day |
| $r_K$ | Residual birth-rate at carrying capacity | 4 ln(2) | Per day |
| $K = K_{D/RT}$ | Carrying capacity | $10^9$ | CFU/g feces |
| $D_0$ | Size of the donor inoculum | $10^7$ | CFU/g feces |
| $N_i$ | Number of distinguishable chromosomal tags | 3 | Dimensionless |
| $N_j$ | Number of distinguishable plasmid tags | 4 | Dimensionless |
| $\mu$ | Re-seeding rate of recipients from systemic sites | Uniform on the discrete grid $10^{-6}$ – $10^3$ in 0.25 log increments | CFU/g feces per day |
| $\gamma = \gamma_{D/T}$ | The rate of conjugation from donors or transconjugants per mating pair | Uniform on the discrete grid $10^{-12}$ – $10^{-3}$ in 0.25 log increments | Per CFU/g feces per day |

## Parameter estimation in the mathematical model

We use ABC (*Marjoram et al., 2003*) to infer the most likely values of migration rate μ and conjugation rate γ. We varied both parameters on a grid, running 100 simulations per parameter set. For each parameter combination, we calculated summary statistics (see below) to compare the simulations to the experimental data.

  i.   The size of the recipient and transconjugant populations on day 10, respectively.
  ii.  The size of the recipient and transconjugant populations on day 10, respectively.
  iii. The time at which the recipient and transconjugant populations first exceed $10^6$ CFU/g feces, respectively.

For *Figure 2C* we calculated the likelihood of the parameter combination as the percentage of simulations that return all summary statistics within three standard deviations of the experimentally observed mean of these statistics. The experimental data used to calculate the mean of these statistics is derived from *Figure 1B*, *Figure 1—figure supplement 1*, and *Figure 2A–B* (n = 8). Due to the assumptions of our model, we excluded mice where re-seeding did not occur. Therefore, the migration rate calculated here may be slightly over-estimated. However, we do not expect the relative importance of re-seeding or conjugation to change due to the exclusion of these mice.

**Appendix 1—table 2.** Parameter estimates.

| Simulation | Maximum likelihood $(\mu,\gamma)$ pair | μ Value from the marginal posterior distribution | γ Value from the marginal posterior distribution |
|---|---|---|---|
| Main text (*Figure 2C*) | $\mu = 1.78$ (CFU/g feces per day) $\gamma = 3.16 \times 10^{-11}$ (per CFU/g feces per day) | 7.1 (CFU/g feces per day) HPD: [0.03, 31.6] | $9.36 \times 10^{-10}$ (per CFU/g feces per day) HPD: [$1 \times 10^{-12}$, $5.6 \times 10^{-9}$] |

HPD: Highest posterior density interval.

## Translation into per-recipient rates

In the mathematical model, the dynamic parameters describing migration and conjugation (μ, $\gamma_{D/T}$) are reported in different units. To facilitate their direct comparison, we can approximate the relative magnitude of these rates per recipient per day.

Recipient migration was inferred to result in 1.78 recipient CFU/g feces per day. This should be divided by the total population of recipients in systemic reservoirs ($8.5 \times 10^5$; *Figure 1C*) to obtain a per-recipient probability of migration. These values result in an approximate probability per recipient (tissue-located) of $2.1 \times 10^{-6}$ per day.

In contrast, γ is a rate constant that needs to be multiplied by two population sizes to be comparable with the migration rate, μ. To obtain a per-recipient (gut-located) conjugation rate reported as CFU/g feces per day, γ needs to be multiplied by the size of the donor population in the gut. Assuming a donor population of $10^9$, this results in conjugation rates of $3.2 \times 10^{-2}$ per day.

## Dependence of transconjugant emergence on the carrying capacity in the gut

In a separate analysis, we estimated the evenness of the plasmid and chromosomal tag distributions, as well as the final recipient and transconjugant population sizes, as a function of the carrying capacity. Here, we fixed the migration rate μ and conjugation rate γ at their most likely values (μ = 1.78 CFU/g feces per day, γ = $3.16 \times 10^{-11}$ per CFU/g feces per day), and varied the carrying capacity K uniformly on the grid $10^3$–$10^{10}$ in 0.25 log increments.

In addition, we repeated this set of simulations with a conjugation rate that was either 100-fold higher, or 100-fold lower than the most likely value, to illustrate a range of plausible plasmid conjugation rates (*Benz et al., 2021*).

