## [Editor Report]

This work reveals an important feature of within-host acquisition and spread of antibiotic resistance. Focusing on a pathogen that shuttles between the gut lumen and tissue reservoirs, this study found that antibiotic degradation in the gut by resistant bacteria can promote the accumulation and spread of plasmids. This manuscript will be of interest to readers in the fields of infection biology, plasmid ecology, gut microbiomes, and antimicrobial resistance.

---

## [Decision Letter]

**Decision letter after peer review:**

Thank you for submitting your article "Pathogen invasion-dependent tissue reservoirs and plasmid-encoded antibiotic degradation boost plasmid spread in the gut" for consideration by *eLife*. Your article has been reviewed by 3 peer reviewers, and the evaluation has been overseen by a Reviewing Editor and Wendy Garrett as the Senior Editor. The reviewers have opted to remain anonymous.

Essential revisions:

1. Since in Figure 1, S1 and S2 recipients only appear after donor inoculation, it is possible that oral inoculation of donors leads to a breach in the gut barrier that in turn facilitates recipient reseeding of the gut lumen. Can the authors provide insight into the mechanism of reseeding of *Salmonella* from the tissue to the gut? Would a TTSS^-1^ off mutant be unable to re-seed the gut lumen and could it serve as a negative control to exclude that re-seeding occurs as an indirect consequence of donor oral inoculation?

2. Figure 3A results need to be strengthened with controls. What happens if the secondary recipient is TTSS^-1^ off, prohibiting conjugation to occur in the tissues? In addition, although unlikely, it does remain formally possible that persisters of the initial *E. coli* donor acted as donor in this second trans-conjugation event. An experiment with tagged plasmids showing that tag distributions in the recipient for the second trans-conjugation event correlate with tag distribution in the initial transconjugant rather than the initial donor might provide a definitive answer.

3. The model used by the authors needs to be explained in a more accessible way. For example, why is recipient migration determined to be the rate limiting step for transconjugant formation? What are the experimental justifications to using 100-fold lower or higher conjugation rates in the model? Why a recipient migration rate of 1.78 events per day is a rate limiting step for transconjugant formation compared with 3.16 x 10^(-11)^ conjugation events per CFU per g feces per day (which superficially seems like a smaller number)? Was a sensitivity analysis for the parameters that were not explored systematically (i.e. for parameters other than mu and γ) performed?

4. Figure 4: in absence of treatment with another antibiotic, please discuss that the results are not, for certain, specific to the β lactamase.

5. Have the authors detected transconjugants by other means than phenotypic antibiotic resistance to show that resistance did not emerge by spontaneous mutation (e.g. PCR or else)?

---

## [Author Response]

Essential revisions:1. Since in Figure 1, S1 and S2 recipients only appear after donor inoculation, it is possible that oral inoculation of donors leads to a breach in the gut barrier that in turn facilitates recipient reseeding of the gut lumen. Can the authors provide insight into the mechanism of reseeding of Salmonella from the tissue to the gut? Would a TTSS^-1^ off mutant be unable to re-seed the gut lumen and could it serve as a negative control to exclude that re-seeding occurs as an indirect consequence of donor oral inoculation?

We agree that the mechanism of re-seeding (and the factors that promote this process) are still not understood. This is explained by the biological features of this process which have precluded detailed mechanistic studies. First of all, re-seeding is a quite rare process that releases no more than a few S.Tm cells into the gut lumen per day. Therefore, the re-seeding even itself is extremely hard to assess by microscopy. Moreover, the routes that cells can traffic from the various tissue reservoirs (the diversity of tissue reservoirs are shown in the current Figure 1—figure supplement 6) to the gut lumen are likely dependent on the nature of the reservoirs the cells are in initially. While we believe that an in-depth investigation of the mechanism(s) of re-seeding remains out of the scope of this work, we have performed control experiments as suggested here to rule out the contribution of donor inoculation, and to confirm that TTSS^-1^ is not essential for re-seeding.

First, while we showed that TTSS-2 encoded on SPI-2 is important for establishing tissue reservoirs (and as a consequence, re-seeding; Figure 1B-C), we additionally tested a TTSS^-1^ mutant (invG mutant). We now show that active invasion because of TTSS^-1^ activity is not essential for either tissue reservoir formation after i.p. infection, or re-seeding from these reservoirs. This is because even with a TTSS^-1^ mutant, cells were found in the same tissue reservoirs as WT S.Tm recipients, and also re-seeded the gut lumen. This data is now presented in Figure 1—figure supplement 3.

Second, we also performed the suggested experiment to investigate a possible contribution of the donor inoculation process. We repeated the same experiment as in Figure 1B-C but split the mice into two groups. In one group the donors were given as usual. In the other group, donors were not given, but the streptomycin treatment was kept as in the first group. Re-seeding occurred in both treatment groups, and both the number of mice (2/4 vs 3/5 mice by day 10 of the experiment) and the population sizes of the re-seeding population were not significantly different. This data is now presented in Figure 1—figure supplement 4. We also tested the diversity of 7 recipient chromosome tags in the populations that were isolated from the feces after re-seeding in both groups: one or few tags dominated the population in both treatment groups. This data is presented in Figure 2—figure supplement 2.

Altogether, these data indicate that re-seeding is likely a rare process determined by trafficking of host cells to the gut and/or the natural circulatory/lymphatic/biliary system of the mouse, rather than an active process dependent on TTSS^-1^ mediated S.Tm invasion. While this may not have a direct impact on re-seeding per se, the most important phenomenon to limit robust growth after re-seeding appears to be colonization resistance conferred by the microbiota (supported by both the low chances for re-seeded luminal populations in the phases of our experiments where CR remains intact and the results of the mathematical model; Figure 2D). It remains possible that external factors like inflammation may boost immune cell recruitment leading to increased re-seeding (in our work, all tested donor strains were not able to induce inflammation, e.g. avirulent *Salmonella* or *E. coli*). Nevertheless, we show that re-seeding can in principle occur in the absence of active invasion (by both donors and recipients).

2. Figure 3A results need to be strengthened with controls. What happens if the secondary recipient is TTSS^-1^ off, prohibiting conjugation to occur in the tissues? In addition, although unlikely, it does remain formally possible that persisters of the initial *E. coli* donor acted as donor in this second trans-conjugation event. An experiment with tagged plasmids showing that tag distributions in the recipient for the second trans-conjugation event correlate with tag distribution in the initial transconjugant rather than the initial donor might provide a definitive answer.

We have done two additional experiments to address these concerns.

First, to provide more evidence that the initial donor (i.e. *E. coli* 536) did not contribute to the secondary trans-conjugation event, we assessed how well it is eliminated by the ciprofloxacin/ampicillin treatment regime (days 10-15). To this end, we performed the same experimental setup as in Figure 3 and the current Figure 3—figure supplement 2. To maximize our chances for detecting any *E. coli* 536 "survivors", we did this experiment in the absence of S.Tm, which should exclude any possible competitive interactions between the two species. We performed the antibiotic treatment protocol as in Figure 3 and the current Figure 3—figure supplement 2 but finished with streptomycin in the drinking water to select for any remaining *E. coli* 536 (which is streptomycin resistant) and against the microbiota. This should further increase the detection-sensitivity for any remaining *E. coli* that could re-seed. We did not detect any *E. coli* in the feces after 3 days of streptomycin in the drinking water, nor in any of the tissue reservoirs. This supported our data in Figure 3—figure supplement 2 suggesting that *E. coli* cannot survive the antibiotic treatment at detectable levels, but also provided further evidence that re-seeding of the initial *E. coli* and a contribution to the secondary plasmid transmission is unlikely. This new data is presented in Figure 3—figure supplement 3.

Second, to directly address if conjugation can in principle happen inside of tissue reservoirs, we performed an experiment in which we introduced both virulent S.Tm donors (SL1344 p2cat) and recipients (14028) directly into the tissue reservoirs by an i.p. infection (both strains introduced by i.p. injection at the same site of the mouse, 5 minutes sequentially). We then allowed them to grow and spread within and between tissue reservoirs for 3 days before euthanizing the mice and assessing transconjugant formation. Importantly, while both donors and recipients were detected at high densities in tissue reservoirs, transconjugants were not detected in any reservoirs. Therefore, this suggests that conjugation is highly unlikely in the absence of interaction of donors and recipients in the gut lumen. This supports that acquisition of the plasmid in Figure 3A is far more likely in the gut lumen after transconjugant re-seeding, rather than invasion and within-tissue plasmid acquisition by the secondary recipient strain. This new data is presented in Figure 3—figure supplement 4.

3. The model used by the authors needs to be explained in a more accessible way. For example, why is recipient migration determined to be the rate limiting step for transconjugant formation? What are the experimental justifications to using 100-fold lower or higher conjugation rates in the model? Why a recipient migration rate of 1.78 events per day is a rate limiting step for transconjugant formation compared with 3.16 x 10^(-11)^ conjugation events per CFU per g feces per day (which superficially seems like a smaller number)? Was a sensitivity analysis for the parameters that were not explored systematically (i.e. for parameters other than mu and γ) performed?

We have added more information to the main and supplementary text to make the model more intuitive and accessible (lines 291-295; 936-947). In particular, we have added a sentence to compare both parameters (mu and γ) in similar units, which helps convey why recipient migration is the rate limiting step (lines 292-294).

In brief, we found recipient migration results in 1.78 recipient CFU/g feces per day. This should be divided by the total population of recipients in systemic reservoirs (see Figure 1C) to obtain a per-recipient probability of migration. These values result in an approximate probability per recipient (tissue-located) of 2.1 x 10^-6^ per day. Instead, conjugation was reported per CFU/g feces per day, and thus must be multiplied by the size of the donor population in the gut to reach a per-recipient (gut-located) probability of conjugation. Assuming the donors are at the carrying capacity of 10^9^, this would result in values around 3 x 10^-2^ per day which is a much greater number.

Our decision to use a 100-fold difference in the conjugation rate in both directions (a total span of 5 orders of magnitude), was based on experience from previous work (Benz et al., 2020, ISME J, https://doi.org/10.1038/s41396-020-00819-4) where we conjugated different IncI plasmids to a range of recipients and found roughly 5 orders of magnitude difference across strains. The chosen conjugation rates are meant to illustrate general trends for more or less conjugative plasmids.

All parameters except mu and γ were parametrised using experimental data, described in greater detail in Bakkeren et al., 2019, Nature. In that study we also reported a scenario with a higher birth rate at carrying capacity (simulating inflammation) and found that it marginally affected the estimated parameters, but did not affect qualitative statements about which steps were rate limiting. We also checked the effect of more general combinations of birth and death rates in this model, and did not find noticeable differences (the dynamics were dominated by conjugation and re-seeding). In the current study we further investigated the effect of the carrying capacity K on the dynamics.

4. Figure 4: in absence of treatment with another antibiotic, please discuss that the results are not, for certain, specific to the β lactamase.

We agree that in-trans protection is best documented for β-lactam antibiotics. As β-lactams alone account for about 2/3 of the annual world-wide antibiotic consumption, the in vivo demonstration of this in trans protection should be of significant interest in itself. We have pointed this out more clearly in the revised text (lines 110-111; 398-399). While we lack direct in vivo evidence that the results relevant to Figure 4 (i.e. cross-protection of susceptible cells by resistant cells leading to HGT) are more general than for β-lactamases, there other plausible mechanisms that could contribute to cross protection that have so far been demonstrated in vitro:

1. The local concentration of antibiotics with an intracellular mechanism of action could also be depleted in the presence of resistant bacteria, depending on the mechanism of resistance. For enzymes that inactivate the antibiotic, high densities of resistant bacteria could deplete the local concentration of antibiotics after internalization. Resistance genes that can inactivate the antibiotic were identified for several classes of antibiotics (D’Costa et al., Science, 2006). It remains to be seen how efficiently these antibiotic-inactivating enzymes would function in vivo. However, this would likely be less efficient compared to secreted enzymes compared to β-lactamases.

2. Some studies have investigated another "in-trans" protection mechanism, i.e. the contribution of outer membrane vesicles in transferring resistance determinants from resistant cells to susceptible cells. In fact, the contribution of such vesicles to the survival of otherwise lethal concentrations of membrane-disrupting antibiotics such as polymyxin B and colistin have been experimentally demonstrated in several different organisms (including *E. coli* and *Salmonella* Typhi; Manning and Kuehn, BMC Microbiol, 2011; Marchant et al., 2021, Front. Microbiol.).

We have expanded the Discussion section (lines 518-533) to encompass these points.

5. Have the authors detected transconjugants by other means than phenotypic antibiotic resistance to show that resistance did not emerge by spontaneous mutation (e.g. PCR or else)?

We have now performed a control PCR on multiple clones from mouse experiments with both recipient chromosome- and plasmid-specific primers to verify that transconjugants are truly recipients that have obtained a plasmid. This data is now presented in Figure 1—figure supplement 2.